# Therapeutic Decision Making in Prevascular Mediastinal Tumors Using CT Radiomics and Clinical Features: Upfront Surgery or Pretreatment Needle Biopsy?

**DOI:** 10.3390/cancers16040773

**Published:** 2024-02-13

**Authors:** Chao-Chun Chang, Chia-Ying Lin, Yi-Sheng Liu, Ying-Yuan Chen, Wei-Li Huang, Wu-Wei Lai, Yi-Ting Yen, Mi-Chia Ma, Yau-Lin Tseng

**Affiliations:** 1Division of Thoracic Surgery, Department of Surgery, National Cheng Kung University Hospital, College of Medicine, National Cheng Kung University, Tainan 704302, Taiwan; n048127@mail.hosp.ncku.edu.tw (C.-C.C.); n511608@mail.hosp.ncku.edu.tw (Y.-Y.C.); n100297@mail.hosp.ncku.edu.tw (W.-L.H.); 071548@tool.caaumed.org.tw (W.-W.L.); tsengyl@mail.ncku.edu.tw (Y.-L.T.); 2Department of Medical Imaging, National Cheng Kung University Hospital, College of Medicine, National Cheng Kung University, Tainan 704302, Taiwan; n049809@mail.hosp.ncku.edu.tw (C.-Y.L.); n041075@mail.hosp.ncku.edu.tw (Y.-S.L.); 3Division of Thoracic Surgery, Department of Surgery, An-Nan Hospital, China Medical University, Tainan 70965, Taiwan; 4Division of Trauma and Acute Care Surgery, Department of Surgery, National Cheng Kung University Hospital, College of Medicine, National Cheng Kung University, Tainan 704302, Taiwan; 5Department of Statistics and Institute of Data Science, National Cheng Kung University, Tainan 701401, Taiwan

**Keywords:** core needle biopsy, prevascular tumors, machine learning, radiomics, surgical resection, voting ensemble, thymoma

## Abstract

**Simple Summary:**

Initial management approaches for prevascular mediastinal tumors (PMTs) can be divided into two categories: direct surgery and core needle biopsy (CNB). Although the gold standard diagnostic method is histopathological examination, the selection of the initial management between direct surgery and CNB is more urgent for patients with PMTs, compared with the definite diagnosis of PMT subtypes. The study aimed to develop clinical–radiomics machine learning (ML) classification models to differentiate patients who needed direct surgery from patients who needed CNB, among the patients with PMTs. An ensemble learning model, combining five ML models, had a classification accuracy of 90.4% (95% CI = 87.9 to 93.0%), which significantly outperformed clinical diagnosis (86.1%; *p* < 0.05), which may be used as clinical decision support system to facilitate the selection of the initial management of PMT.

**Abstract:**

The study aimed to develop machine learning (ML) classification models for differentiating patients who needed direct surgery from patients who needed core needle biopsy among patients with prevascular mediastinal tumor (PMT). Patients with PMT who received a contrast-enhanced computed tomography (CECT) scan and initial management for PMT between January 2010 and December 2020 were included in this retrospective study. Fourteen ML algorithms were used to construct candidate classification models via the voting ensemble approach, based on preoperative clinical data and radiomic features extracted from the CECT. The classification accuracy of clinical diagnosis was 86.1%. The first ensemble learning model was built by randomly choosing seven ML models from a set of fourteen ML models and had a classification accuracy of 88.0% (95% CI = 85.8 to 90.3%). The second ensemble learning model was the combination of five ML models, including NeuralNetFastAI, NeuralNetTorch, RandomForest with Entropy, RandomForest with Gini, and XGBoost, and had a classification accuracy of 90.4% (95% CI = 87.9 to 93.0%), which significantly outperformed clinical diagnosis (*p* < 0.05). Due to the superior performance, the voting ensemble learning clinical–radiomic classification model may be used as a clinical decision support system to facilitate the selection of the initial management of PMT.

## 1. Introduction

The mediastinum, located in the center of the thoracic cavity, separates the left and right pleural cavities. The International Thymic Malignancy Interest Group (ITMIG) system divides the mediastinum into prevascular, visceral, and paravertebral compartments [1]. Several large-scale computed tomography (CT) screening studies have revealed that the prevalence of prevascular mediastinal tumors (PMTs) varies from 0.73% to 1.49% [2,3,4]. The prevascular mediastinal compartment harbors structures such as the thymus, fat, lymph nodes, and the left brachiocephalic vein [5]. Depending on their origin within specific regions, PMTs comprise different subtypes, each with distinct pathologic and radiologic characteristics, such as thymic epithelial tumors, cysts, thymic hyperplasia, teratomas, goiters, lymphomas, and malignant germ cell tumors [6,7,8].

Given that PMTs are highly heterogeneous diseases, it is imperative to tailor different management strategies to the specific subtypes [9,10]. For example, guidelines recommend medical treatment rather than surgical resection for lymphomas [11]. In contrast, resectable thymic epithelial tumors (TETs) require surgical resection to avoid the risk of tumor spread during a biopsy of an encapsulated thymoma [12]. Initial management approaches for PMTs can be divided into two categories: direct surgery and core needle biopsy (CNB). Category 1 includes patients with resectable TETs, cysts, thymic hyperplasia, teratomas, goiters, lymphangiomas, or thymolipomas, who are advised to undergo immediate surgical resection [13,14,15]. Category 2 includes patients with unresectable TETs, lymphomas, Castleman disease, or malignant germ cell tumors, who are recommended to start with CNB [15,16,17]. After a confirmed pathologic diagnosis, Category 2 patients may receive targeted therapy, chemotherapy, and/or radiotherapy [18,19].

Clinicians take age, preoperative clinical characteristics, and preoperative radiological features into consideration when making a clinical diagnosis. Thymic epithelial tumors are more common in adults aged 50–70 years; lymphoma and malignant germ cell tumors are more common in adults aged 20–40 years. Patients with malignant germ cell tumor often have higher serum levels of alpha-fetoprotein (AFP) and human chorionic gonadotropin (HCG). Regarding radiological features, benign teratoma often has a fat component, lymphoma often has multiple mediastinal lymphadenopathy, resectable thymoma often has a rounder shape with regular margin, and unresectable thymoma and thymic carcinoma often have great vascular encasement. Clinicians then recommend the suitable initial management based on the clinical diagnosis. The accuracy of such binary classification is validated by the final pathology report, and misclassification results in unnecessary CNB and surgical resection with a negative impact on patient health.

With the advancement of machine learning (ML) technology, various image-based ML models have been developed to predict breast cancer metastasis [20], lung diseases [21,22], and ductal carcinoma [23]. Furthermore, radiomics-based predictive models have been developed to differentiate between PMT subtypes [24,25]. Liu et al. [26] reported that a radiomics-based ML model for differentiating anterior mediastinal cysts from thymomas had a sensitivity of 89.3%, while the sensitivity of the ML model could be improved to 92.3% if both clinical and radiological features were used to construct an ML model. Several clinical–radiomics models have been developed to predict pathological subtypes of PMTs [27], to differentiate low-risk from high-risk thymomas [28] and to distingue TETs from other PMT subtypes [29], thereby facilitating clinical diagnosis. However, no clinical–radiomics models have been developed to predict the selection of the initial management between direct surgery and CNB.

The gold standard diagnostic method for PMTs is histopathological examination. Compared to the definite diagnosis of PMT subtypes, the selection of the initial management between direct surgery and CNB is a more urgent issue that must be solved quickly in clinical practice. Imaging findings and clinical characteristics may be useful to differentiate surgical cases from nonsurgical cases [30]. Therefore, this retrospective study aimed to develop ML classification models using preoperative clinical features and radiomics to differentiate Category 1 patients who need immediate surgical resection from Category 2 patients who need CNB first. These expected classification models will contribute to the development of an ML-based clinical decision support system to facilitate the selection of the appropriate initial management for each individual patient with PMT.

## 2. Methods

### 2.1. Patient Selection and Classification

Consecutive patients aged 20 years or older and with PMT, who underwent surgical resection or CNB between January 2010 and December 2020, were initially selected. We enrolled patients who underwent a complete contrast-enhanced computed tomography (CECT) of the chest. Patients who underwent non-contrast-enhanced computed tomography alone or only had head and neck computed tomography without complete coverage of the entire thoracic cavity were excluded. Eligible patients were divided into two categories based on the final pathology reports. Category 1 patients had pathologically confirmed resectable TET, thymic hyperplasia, cyst, lymphangioma, thymolipoma, or teratoma and would undergo direct surgery. Category 2 patients had pathologically confirmed unresectable TET, lymphoma, Castleman disease, or malignant germ cell tumor and would receive CNB first. The protocol of this retrospective study was reviewed and approved by the Institutional Review Board (IRB) of the National Cheng Kung University Hospital (IRB No. A-ER-111-287; 4 October 2022), and the requirement for informed consent was waived due to the retrospective nature of this study.

### 2.2. Clinical Diagnosis and Clinical Data Collection

From January 2010 to December 2020, more than 20 clinicians specializing in thoracic surgery, thoracic medicine, oncology, and radiology, participated in the clinical diagnosis. Clinical diagnosis and the subsequent treatment selection were based on the baseline clinical data and preoperative CECT.

A retrospective chart review was performed to collect patient demographic and baseline clinical characteristics, such as age, sex, presence of myasthenia gravis (MG) symptoms, pleural effusion or mediastinal lymphadenopathy (LAP) on CECT, and the serum levels of tumor markers including AFP, HCG, and lactate dehydrogenase (LDH). Serum tumor marker levels were categorized as normal, higher, or missing. It is worth noting that tumor markers are not part of routine blood tests, and clinicians may order them for patients with suspected malignant germ cell tumor.

### 2.3. Image Acquisition and Preprocessing

Four CT scanners, including a Siemens SOMATOM Definition Flash, Siemens SOMATOM Definition AS, Siemens SOMATOM Sensation 16, and GE Optima CT660, were used to acquire CECT images. Contrast medium (60 to 120 mL) was intravenously administered at a rate of 1.5 mL/s, followed by a 20 mL saline flush. All CECT images were obtained 90 s after contrast medium administration. The image size was 512 × 512 pixels. All images were reconstructed in 5 mm slices with a smooth standard convolution kernel (B40f), as previously described [31]. After CECT images were imported into the open-source software 3D Slicer version 4.10.2, the PMTs were manually contoured by a thoracic radiologist (C.Y.L.) with 9 years of experience, blinded to patient diagnosis, using the built-in paint tool as previously described [32]. Tumor segmentation was performed in the mediastinal setting (window level, 50 HU; window width, 350 HU) on the axial CT plane. For normalization, all CT voxels were resampled to 1 mm^3^ using a cubic interpolation.

### 2.4. Radiomic Feature Extraction and Selection

The open-source platform PyRadiomics was used to extract 3D radiomic features from the segmentation of the PMTs on the images [33]. A total of 851 radiomic features were extracted, including 14 shape features, 18 intensity histogram features, 74 texture features, and 745 wavelet features. The least absolute shrinkage and selection operator (LASSO) regression analysis is a feature selection method based on a linear regression model. Both the 851 radiomic features and all the above clinical data were entered into the LASSO regression for variable selection, as previously described [28,29]. The extracted radiomic features reflect subtle characteristics of MPTs in images, such as the sphericity, diameter, homogeneity, and calcification. The variables with non-zero coefficients and the optimal lambda value were selected by the LASSO regression to build ML classification models to distinguish Category 1 patients from Category 2 patients.

### 2.5. Machine Learning Model Building

Python version 3.8.9 with AutoGluon Tubular classifier version 0.8.2 [34] was used to model 14 default ML algorithms, including CatBoost, ExtraTrees with Entropy, ExtraTrees with Gini, Kneighbors with Distance Weights, Kneighbors with Uniform Weights, LightGBM, LightGBMLarge, LightGBM with ExtraTress, NeurlNetFastAI, NeuralNetTorch, RandomForest with Entropy, RandomForest with Gini, WeightedEnsemble_L2, and XGBoost to develop the classification models.

Three different approaches were used to develop ML classification models to discriminate between Category 1 patients and Category 2 patients. First, all clinical variables and radiomic features were used to train and validate 14 different ML models using 10-fold cross validation, a common approach for evaluating the performance of ML prediction models [35,36,37]. The dataset was divided into 10 subsets. The model was trained on 9 randomly selected subsets and was validated on the remaining subset. The model was then trained and validated in the same way 10 times. Thus, the macro F1 score, macro precision, macro recall, accuracy, and area under the receiver operating characteristic curve (AUROC) calculated from each time were averaged to estimate the generalization performance of the model, as previously described [38].

Additionally, LASSO regression was applied to all clinical variables and radiomic features, resulting in 20 combinations of the selected variables based on various lambda values. These combinations are labeled as Selection_1 to Selection_20. The features selected by LASSO selection, as well as all features without LASSO selection, were independently used to build the aforementioned models.

Subsequently, we constructed voting ensemble ML models based using Python version 3.8.9. The final decision was made by majority voting. The features selected by Selection_3 and all features without LASSO selection were used independently to build voting ensemble learning models. The performance of all combinations of 3, 5, 7, or 9 ML models, which were randomly selected from a total of 14 ML models, was evaluated. Two voting ensemble approaches were used to select the optimal ML models. First, the mean classification performances of these combinations of 3, 5, 7, or 9 were averaged separately. The combination of a given number of ML models with the highest mean accuracy was identified as the final model. Second, the single combined ML model with the highest accuracy was identified as the optimal voting ensemble learning model. The construction workflow of the voting ensemble learning models is shown in Figure 1.

### 2.6. Statistical Analyses

Age is expressed as the median (Q1, Q3), and between-group difference was examined using the Mann–Whitney U test. Categorical variables are expressed as counts (percentages), and between-group differences were compared using the chi-square and Fisher’s exact test. As 10-fold cross-validation was used in this study, the macro F1 score, macro precision, macro recall, accuracy, and AUROC were expressed as the mean ± SD. Statistical significance was set at a *p* value of 0.05 or 95% conference interval (95% CI) [39]. The *p* value represents the probability of committing a Type I error, while the significance level denotes the upper limit of the acceptable Type I error probability, typically set at 0.05, 0.01, or 0.001. The choice of the significance level depends on the researcher’s willingness to take on the risk of making a decisional error. Setting a significance level of 0.05 means allowing the possibility (or probability) that a false alarm will occur, which should be less than 0.05 (i.e., only 1 occurrence in 20). Statistical analysis of the ML models was performed using statsmodels version 0.13.1.

## 3. Results

### 3.1. Patient Selection and Grouping

A total of 375 eligible patients were included in the study. According to the final pathology reports, 182 patients were to undergo direct surgery (Category 1), and 193 patients were to undergo CNB first (Category 2). The flowchart of the patient selection and grouping is shown in Figure 2.

### 3.2. Baseline Demographic and Clinical Characteristics

The baseline demographic and clinical characteristics between the two categories are presented in Table 1. The mean age of Category 1 patients was significantly older than that of Category 2 patients (*p* < 0.0001). Category 1 patients had a significantly higher percentage of MG symptoms but lower percentages of pleural effusion and mediastinal lymphadenopathy than Category 2 patients (all *p* < 0.0001). After excluding patients with missing data, Category 2 had significantly more patients with higher levels of LDH and AFP than Category 1 (both *p* ≤ 0.0147). There were no significant differences in sex and serum HCG between the two categories (Table 1). The centrality and dispersion for age, LDH, AFP, and HCG are presented in Appendix A.

### 3.3. Classification Accuracy for Clinical Diagnosis

With reference to the final pathology reports (gold standard), the classification accuracy of the clinical diagnosis was 86.1% (323/375). This means that 52 out of 375 patients were wrongly classified by the clinical diagnosis in the first place. Of these, 15 patients were misclassified by clinical diagnosis into Category 2, and 37 patients were misclassified by clinical diagnosis into Category 1.

### 3.4. The Individual Machine Learning Model

Among 14 different ML models, the classification model based on the CatBoost algorithm had the best classification performance, with an accuracy of 0.8227 ± 0.0430 (Table 2). However, the accuracy of the CatBoost model was 82.27 (95% CI = 79.6 to 84.9%), which was significantly lower than that of clinical diagnosis (86.1%), because 86.1% did not fall within the 95% CI of the CatBoost model.

The CatBoost model with the third LASSO selection (Selection_3) had the best classification performance, with an optimal lambda of 0.025354, a Ln (lambda) of −3.675, and an accuracy of 0.8422 ± 0.0423 (Table 3). Therefore, the variables selected by Selection_3 were subsequently used for classification modeling (Appendix A). Nevertheless, the accuracy of the CatBoost model with Selection_3 was 84.2% (95% CI = 81.6 to 86.8%), which was still significantly lower than that of the clinical diagnosis (86.1%).

### 3.5. Voting Ensemble Machine Learning Models

The results showed that the random selection of seven ML models from a total of fourteen ML models using the features selected by Selection_3 had the best average classification performance, with an accuracy of 0.8804 ± 0.0358 (Table 4). The accuracy of this combination ML was 88.0% (95% CI = 85.8 to 90.3%), which was not significantly different from that of the clinical diagnosis (86.1%).

In addition, the classification performance of all possible combinations of three, five, seven, or nine ML models using the features selected by Selection_3 were evaluated by voting process. The best combination with the highest classification accuracy was a combination of five ML models, including NeuralNetFastAI, NeuralNetTorch, RandomForest with Entropy, RandomForest with Gini, and XGBoost, with an accuracy of 0.9044 ± 0.0408. The accuracy of this ensemble learning classification model was 90.4% (95% CI = 87.9 to 93.0%), which was significantly higher than that of the clinical diagnosis (86.1%) (*p* < 0.05).

## 4. Discussion

This retrospective study revealed that the clinical diagnosis had an accuracy of 86.1%. Of the 14 ML algorithms evaluated, the CatBoost model achieved the highest accuracy of 84.2%. The first voting ensemble ML model, which randomly selected seven ML models from the set of fourteen, had an average classification accuracy of 88.0%. The second voting ensemble ML model, a specific combination of five ML models, achieved an accuracy of 90.4%. In particular, the classification accuracy of the second voting ensemble ML model was significantly higher than that of the clinical diagnosis.

Although the gold standard diagnostic method is histopathological examination, the initial management for patients with PMT is determined based on clinical diagnosis but not pathological diagnosis, indicating the importance of clinical diagnostic tests [40]. In clinical practice, a multidisciplinary team works together to reach a consensus on the clinical diagnosis and initial management for patients with PMT. This consensus-based decision-making process is time- and labor-consuming. Moreover, a discrepancy in clinical diagnostic suggestions might occur in patients aged 30–40 or older than 70, who have normal levels of tumor markers but no specific radiological features. Therefore, ML classification models with superior classification accuracy may serve as clinical decision support systems for differential diagnosis and management. Notably, radiomics-based ML models have been developed to predict overall survival in various cancer sites [41] and to predict management in lower-grade gliomas [42].

Accurate clinical diagnosis is crucial for making treatment decisions, especially for highly diverse PMTs [6,10]. While imaging findings have been widely used to differentiate PMT subtypes based on location, fat content, and calcification [8,30,43], biopsies are necessary when imaging results are inconclusive. CT-guided percutaneous CNB has been shown to be effective and safe for the diagnosis of PMTs, with diagnostic yields ranging from 75.7% to 96% across studies [16,44,45].

In this retrospective study, CECT images were generated by four different CT scanners. Because PMTs are rare and heterogeneous tumors, we were not able to evaluate the influence of different CT scanners on image quality and characteristics in patients with the same PMT subtype. Notably, the use of different CT scanners does not affect the performance of ML models for detecting lung nodules [46], but the reconstruction kernel affects radiomic feature selection and model performance [31]. Thus, in this study, all CECT raw data were reconstructed with a smooth standard convolution kernel (B40f) to reduce variations in image characteristics.

Automated extraction of a huge amount of quantitative radiomic features allows analyzing subtle characteristics of tumors on images, such as the sphericity, diameter, homogeneity, and calcification, thereby facilitating disease differential diagnosis [47]. Resectable TETs usually have regular and smooth borders and smaller diameters, while lymphomas, malignant germ cell tumors, and unresectable TETs tend to have larger diameters and irregular borders. Lymphoma is more homogenous, but TET and teratoma are more heterogeneous and may be calcified, resulting in differences in the Hounsfield Unit value. 

Notably, only one clinical feature, the serum level of LDH, was extracted by LASSO in Selection_3. Although patients with a malignant germ cell tumor often have higher levels of AFP and HCG, those clinical features with high specificity were not selected by LASSO, which may in part be due to few patients having a malignant germ cell tumor in the study population. Although LDH has a low specificity for differential diagnosis of PMT subtype, PMTs with higher malignancy and fast growth, such as lymphoma, high-grade thymoma, thymic carcinoma, and malignant germ cell tumor, usually have higher LDH levels. Therefore, serum LDH may be considered for inclusion in routine preoperative blood tests. In support of this suggestion, an elevated serum LDH level has been reported in conditions such as mediastinal seminoma, mediastinal lymphoma, and leukocytosis [48].

The ML algorithms used in this study have been commonly utilized to develop classification models for various diseases [49,50,51,52,53]. Ensemble learning techniques have been widely used in clinical practice to combine several ML models to create a stronger model with superior performance. The ensemble model has a better predictive accuracy than those of individual models [50]. RandomForest, CatBoost, and XGBoost were used to develop an ensemble model for predicting pre-cancer in pre- and post-menopausal women, with an accuracy of 94% [51]. In particular, voting ensemble methods have been used to improve the performance of prediction models for the detection of cervical cancer and brain tumors [54,55], prognosis of non-small cell lung cancer [53], and the diagnosis and prognosis of acute coronary syndrome [56]. In our research, we applied voting ensemble methods to construct an ensemble learning model with a classification accuracy of 90.4%, which significantly outperformed the clinical diagnosis.

Deep learning utilizes artificial neural networks to simulate the human brain. Generally speaking, deep learning models outperformed the ML models; however, deep learning models are more complex and require more parameters, huge amounts of data, and high computational cost. In our recent study, we found that the radiomics-based ML model had superior performance compared to the 3D conventional neural network (CNN) model within a limited dataset [29]. Due to the limited sample size, the present study focused on the development of ML models using both clinical features and radiomics.

Compared with other clinical–radiomics model studies that limited specific PMT histology types [27,28,29], this study did not set restrictions on PMT subtypes, including 12 different PMT subtypes. In addition, unlike other clinical–radiomics model studies aimed to improve differential diagnosis [27,28,29], the developed voting ensemble ML model may be utilized as a clinical decision support system to help the selection of initial management for patients with PMTs.

Some limitations need to be addressed. First of all, the findings of this single-institution retrospective study need to be confirmed by other studies conducted in distinct geographic areas. In addition, this retrospective study did not have a training dataset and a separated validation dataset because of a spectrum of many PMT subtypes. Instead, 10-fold cross validation was utilized to train and validate ML models. The mean classification failure rates of ensemble learning models could be calculated in the present retrospective study; however, patients at high risk of misclassification and the underlying reasons could not be investigated. Moreover, due to the etiologic diversity of PMTs [6,57], many PMT subtypes are extremely rare and may not be considered in the binary classification system for initial treatment. Therefore, large-scale multicenter studies are warranted to evaluate the feasibility of ensemble ML classification model as a clinical decision support system to facilitate the selection of initial treatment for patients with PMTs.

Furthermore, some unresectable TET cases may receive neoadjuvant chemotherapy first. Once the tumor size is reduced, surgical resection will be performed to improve proportion of R0 resection and the prognosis. However, due to the small number of patients undergoing neoadjuvant chemotherapy followed by surgery, large-scale multicenter studies are needed to develop a reliable model for predicting patients with TET who may undergo neoadjuvant chemotherapy followed by surgical resection.

## 5. Conclusions

In conclusion, based on preoperative clinical and radiomic features, we developed a voting ensemble learning classification model to discriminate patients with PMT who need direct surgery from those who need CNB, achieving a classification accuracy of 90.4% that was higher than the clinical diagnosis (86.1%). This voting ensemble learning clinical–radiomic classification model can serve as a clinical decision support system to assist clinicians in selecting the appropriate initial treatment for PMT patients to reduce unnecessary CNB or surgical resection and the harm to patients.

## Figures and Tables

**Figure 1 cancers-16-00773-f001:**
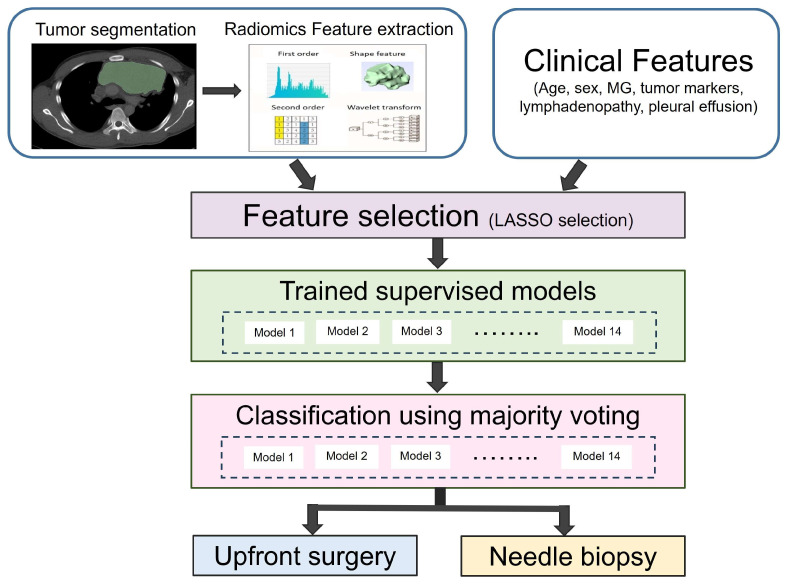
The construction workflow of voting ensemble learning models.

**Figure 2 cancers-16-00773-f002:**
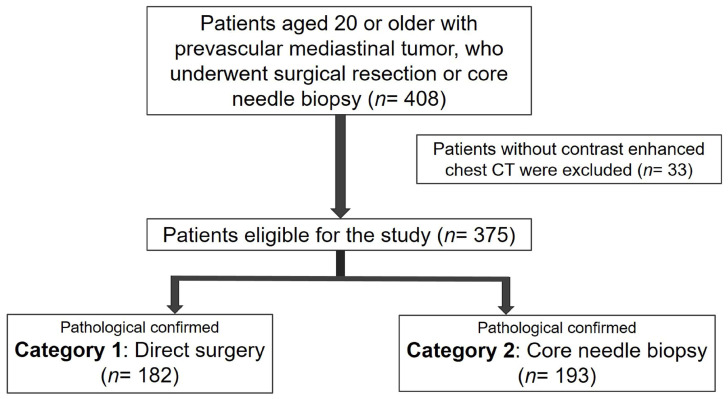
Flowchart of patient selection and grouping based on the final pathology reports.

**Table 1 cancers-16-00773-t001:** Baseline demographic and clinical characteristics.

Characteristics	Total Population (*n* = 375)	Category 1 (*n* =182)	Category 2 (*n* = 193)	*p*-Value
Sex				0.6387
Female	189 (50.4)	94 (51.7)	95 (49.2)	
Male	186 (49.6)	88 (48.3)	98 (50.8)	
Age, median (Q1, Q3)	59 (42, 69)	61 (53, 70)	54 (33, 68.5)	**<0.0001**
MG symptoms	57 (15.2)	49 (26.9)	8 (4.1)	**<0.0001**
Pleural effusion	87 (23.2)	18 (9.9)	69 (35.8)	**<0.0001**
Mediastinal lymphadenopathy	181 (48.3)	39 (21.4)	122 (63.2)	**<0.0001**
Tumor markers				
^†^ LDH (U/L)	147 (39.2)	31	116	
Normal (≤225)	68	22	46	**0.0019**
Higher (>225)	79	9	70	
^†^ AFP (mg/L)	127 (33.9)	45	82	
Normal (≤20)	117	45	72	**0.0147**
Higher (>20)	10	0	10	
^†^ HCG (IU/L)	99 (26.4)	31	68	
Normal (≤7)	92	31	61	0.0639
Higher (>7)	7	0	7	
Diagnosis				
Resectable thymoma		109		
Resectable thymic carcinoma		20		
Thymic hyperplasia		1		
Cyst		32		
Teratoma		14		
Thymolipoma		5		
Lymphangioma		1		
Unresectable thymoma			23	
Unresectable thymic carcinoma			77	
Lymphoma			76	
Malignant germ cell tumor			16	
Castleman disease			1	

Age is reported as the median (Q1, Q3). The other characteristics are presented as the number of patients and percentage, n (%). **^†^** 228, 248, and 276 patients had missing data on LDH, AFP, and HCG, respectively. Abbreviations: LDH: lactate dehydrogenase; AFP: alpha fetoprotein AFP; HCG: human chorionic gonadotropin; the bold font indicates statistical significance (*p* < 0.05).

**Table 2 cancers-16-00773-t002:** Performance of the 14 ML classification models.

Algorithms	Macro F1-Score	Macro Precision	Macro Recall	Accuracy	AUROC
**CatBoost**	**0.8222 ± 0.0433**	**0.8253 ± 0.0428**	**0.8231 ± 0.0431**	**0.8227 ± 0.0430**	**0.8937 ± 0.0402**
ExtraTrees with Entropy	0.7700 ± 0.0468	0.7717± 0.0465	0.7704 ± 0.0467	0.7707 ± 0.0465	0.8658 ± 0.0405
ExtraTrees with Gini	0.7637 ± 0.0458	0.7658 ± 0.0455	0.7642 ± 0.0458	0.7644 ± 0.0454	0.8628 ± 0.0393
Kneighbors with Distance Weights	0.6906 ± 0.0521	0.6940 ± 0.0501	0.6916 ± 0.0508	0.6924 ± 0.0511	0.7615 ± 0.0456
Kneighbors with Uniform Weights	0.6910 ± 0.0510	0.6945 ± 0.0488	0.6920 ± 0.0497	0.6929 ± 0.0499	0.7562 ± 0.0435
LightGBM	0.8040 ± 0.0408	0.8064 ± 0.0412	0.8048 ± 0.0412	0.8044 ± 0.0406	0.8792 ± 0.0379
LightGBMLarge	0.8131 ± 0.0434	0.8180 ± 0.0401	0.8141 ± 0.0429	0.8142 ± 0.0422	0.8962 ± 0.0318
LightGBM with ExtraTrees	0.8203 ± 0.0453	0.8231 ± 0.0444	0.8210 ± 0.0451	0.8209 ± 0.0449	0.8939 ± 0.0419
NeuralNetFastAI	0.7314 ± 0.0669	0.7472 ± 0.0578	0.7379 ± 0.0618	0.7342 ± 0.0634	0.8272 ± 0.0495
NeuralNetTorch	0.7837 ± 0.0525	0.7878 ± 0.0534	0.7848 ± 0.0525	0.7844 ± 0.0523	0.8658 ± 0.0475
RandomForest with Entropy	0.8036 ± 0.0456	0.8059 ± 0.0450	0.8045 ± 0.0455	0.8040 ± 0.0454	0.8779 ± 0.0393
RandomForest with Gini	0.8013 ± 0.0475	0.8038 ± 0.0465	0.8024 ± 0.0474	0.8018 ± 0.0472	0.8722 ± 0.0420
WeightedEnsemble_L2	0.8128 ± 0.0459	0.8156 ± 0.0451	0.8136 ± 0.0457	0.8133 ± 0.0457	0.8901 ± 0.034
XGBoost	0.8186 ± 0.0420	0.8215 ± 0.0419	0.8194 ± 0.0422	0.8191 ± 0.0417	0.8927 ± 0.0337

Best results are in bold.

**Table 3 cancers-16-00773-t003:** Classification performance of the Catboost model with different combinations of selected features or all features without LASSO selection.

	Lambda	Macro F1-Score	Macro Precision	Macro Recall	Accuracy	ROC-AUC
Selection_1	0.048626	0.8183 ± 0.0441	0.8220 ± 0.0431	0.8188 ± 0.0443	0.8191 ± 0.0435	0.9061 ± 0.0345
Selection_2	0.035112	0.8306 ± 0.0350	0.8327 ± 0.0348	0.8308 ± 0.0350	0.8311 ± 0.0347	0.9106 ± 0.0301
**Selection_3**	**0.025354**	**0.8417 ± 0.0426**	**0.8444 ± 0.0423**	**0.8422 ± 0.0427**	**0.8422 ± 0.0423**	**0.9114 ± 0.0324**
Selection_4	0.018307	0.8305 ± 0.0447	0.8327 ± 0.0448	0.8307 ± 0.0447	0.8311 ± 0.0445	0.9095 ± 0.0356
Selection_5	0.013219	0.8300 ± 0.0471	0.8325 ± 0.0471	0.8302 ± 0.0471	0.8307 ± 0.0467	0.9092 ± 0.0352
Selection_6	0.009545	0.8264 ± 0.0380	0.8296 ± 0.0387	0.8266 ± 0.0380	0.8271 ± 0.0378	0.9027 ± 0.0350
Selection_7	0.006893	0.8305 ± 0.0499	0.8334 ± 0.0491	0.8309 ± 0.0498	0.8311 ± 0.0496	0.8971 ± 0.0450
Selection_8	0.004977	0.8385 ± 0.0329	0.8419 ± 0.0326	0.8389 ± 0.0329	0.8391 ± 0.0326	0.9064 ± 0.0301
Selection_9	0.003594	0.8347 ± 0.0366	0.8372 ± 0.0371	0.8353 ± 0.0363	0.8351 ± 0.0366	0.9086 ± 0.0304
Selection_10	0.002595	0.8365 ± 0.0465	0.8391 ± 0.0466	0.8374 ± 0.0463	0.8369 ± 0.0464	0.9030 ± 0.0423
Selection_11	0.001874	0.8315 ± 0.0436	0.8346 ± 0.0429	0.8325 ± 0.0436	0.8320 ± 0.0433	0.9025 ± 0.0293
Selection_12	0.001353	0.8315 ± 0.0439	0.8344 ± 0.0448	0.8318 ± 0.0437	0.8320 ± 0.0440	0.9023 ± 0.0408
Selection_13	0.000977	0.8274 ± 0.0426	0.8303 ± 0.0436	0.8278 ± 0.0427	0.8280 ± 0.0425	0.9034 ± 0.0375
Selection_14	0.000705	0.8319 ± 0.0362	0.8349 ± 0.0366	0.8324 ± 0.0361	0.8324 ± 0.0360	0.9066 ± 0.0336
Selection_15	0.000509	0.8275 ± 0.0479	0.8304 ± 0.0481	0.8281 ± 0.0481	0.8280 ± 0.0478	0.8978 ± 0.0341
Selection_16	0.000368	0.8284 ± 0.0419	0.8303 ± 0.0420	0.8286 ± 0.0419	0.8289 ± 0.0417	0.8981 ± 0.0430
Selection_17	0.000266	0.8311 ± 0.0443	0.8335 ± 0.0445	0.8316 ± 0.0442	0.8316 ± 0.0442	0.9016 ± 0.0408
Selection_18	0.000192	0.8319 ± 0.0407	0.8348 ± 0.0402	0.8326 ± 0.0406	0.8324 ± 0.0405	0.9034 ± 0.0353
Selection_19	0.000138	0.8352 ± 0.0438	0.8377 ± 0.0447	0.8357 ± 0.0436	0.8356 ± 0.0438	0.9030 ± 0.0422
Selection_20	0.000100	0.8238 ± 0.0636	0.8263 ± 0.0626	0.8243 ± 0.0632	0.8244 ± 0.0630	0.8920 ± 0.0497
All	-	0.8222 ± 0.0433	0.8253 ± 0.0428	0.8231 ± 0.0431	0.8227 ± 0.0430	0.8937 ± 0.0402

Best results are in bold.

**Table 4 cancers-16-00773-t004:** Summary of results of voting ensemble.

LASSO Selection	Model Count	Macro F1-Score	Macro Precision	Macro Recall	Accuracy	ROC-AUC
Selection_3	3	0.8606 ± 0.0580	0.8620 ± 0.0567	0.8638 ± 0.0579	0.8614 ± 0.0577	0.8977 ± 0.0476
Selection_3	5	0.8530 ± 0.0597	0.8618 ± 0.0503	0.8542 ± 0.0587	0.8561 ± 0.0565	0.9170 ± 0.0368
**Selection_3**	**7**	**0.8799 ± 0.0358**	**0.8800 ± 0.0355**	**0.8831 ± 0.0358**	**0.8804 ± 0.0358**	**0.9250 ± 0.0353**
Selection_3	9	0.8790 ± 0.0361	0.8795 ± 0.0358	0.8810 ± 0.0366	0.8798 ± 0.0362	0.9296 ± 0.0343
All	3	0.8229 ± 0.0494	0.8248 ± 0.0490	0.8257 ± 0.0488	0.8239 ± 0.0492	0.8706 ± 0.0390
All	5	0.8094 ± 0.0423	0.8176 ± 0.0427	0.8095 ± 0.0423	0.8128 ± 0.0416	0.8872 ± 0.0338
All	7	0.8341 ± 0.0412	0.8358 ± 0.0407	0.8369 ± 0.0404	0.8350 ± 0.0412	0.8950 ± 0.0310
All	9	0.8254 ± 0.0396	0.8285 ± 0.0406	0.8265 ± 0.0393	0.8271 ± 0.0401	0.8995 ± 0.0294

Best results are in bold.

## Data Availability

The data are available from the corresponding author (Y.T.Y), upon reasonable request.

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
