# Peer review of "Therapeutic Decision Making in Prevascular Mediastinal Tumors Using CT Radiomics and Clinical Features: Upfront Surgery or Pretreatment Needle Biopsy?"

_cancers, 2024, doi:10.3390/cancers16040773_

Round 1
Reviewer 1 Report
Comments and Suggestions for Authors
The title is too long, and the essential information is lost.
Simple Summary is missing
Abstract
- Write the aim at the past tense.
- Provide the 95% CI for accuracy.
- The methods are not clear.
- The results are not sufficiently specific.
Introduction
- "In clinical practice, clinicians predict PMT subtypes based on preoperative clinical and radio-logic indicators and make corresponding recommendations for the initial management" please clarify the process.
- "misclassifications can occur" how often?
- "extensively" did not fit with two references.
- "combined predictive models using both clinical and radiological features have been constructed to improve the differential diagnostic performance" briefly present them, along with advantages and disadvantages. The need of a new method must be appropriately substantiated.
- Write the aim of the study at the past tense.
Methods
- "How-ever, patients without complete contrast-enhanced computed tomography (CECT)". What about incomplete "pre-operative clinical features" data ?
- "Clinical diagnosis and the subsequent treatment selection were based on baseline clinical data and preoperative CECT." the gold standard diagnostic in medicine is histopathological examination. Without such an exam the accuracy of any model is questionable.
- "It's" should be read as "It is"
- "Intravenous contrast was administered (injection dose 60-120 mL at a rate of 1.5 ml/s) was performed" please rephrase for clarity.
- "3D radiomic" do not start a sentence with an abbreviation.
- how radiomic features were selected?
- How the algorithms were selected?
- "Continuous variables are presented as mean standard deviation" regardless the distribution?
- It is unclear if you have or not a training and a validation set.
Results
- In the flow something is wrong ... 408 eligible ... 375 without contrast CT scan and 375 included in the study.
- A result must stay either in text or in table/figure (see Figure 3).
- "In the present study, three different approaches were used to develop ML classifica-tion models to discriminate between Category 1 patients and Category 2 patients. First, all clinical variables and radiomic features were used to train 14 different ML models us-ing 10-fold cross validation." duplicate information that belngs to the Methods section; please delete.
- "All clinical variables and radiomic features were then subjected to LASSO regression 20 times, resulting in 20 different combinations of selected variables, designated as Selec-tion_1 to Selection_20. The features selected by each run of LASSO selection, as well as allfeatures without LASSO selection, were independently used to build the CatBoost model, in order to further improve the classification performance of the CatBoost model." this information belongs to the Methods section.
- "Voting ensemble learning models were also built. The features selected by Selec-tion_3 and all features without LASSO selection were used independently to build voting ensemble learning models. The classification performances of all combinations of 3, 5, 7, or 9 ML models were then separately averaged." Methods
- The Results section is a mixture of Methods and Results, sometimes with results duplicated in text and tables/images.
Discussion
- Do not write again the results.
- "We used 10-fold cross-validation, a common approach for evaluating the perfor-mance of ML prediction models [44-46]. The dataset was divided into 10 subsets, and the model was trained and validated 10 times. Each time, the model was trained on 9 randomly selected subsets, and was validated on the remaining subset. Thus, the macro F1 score, macro precision, macro recall, accuracy, and AUROC calculated from each time were averaged to estimate the generalization performance of the model." this information belongs to Methods section.
- Begin the discussion by briefly summarizing the main findings.
- Explore possible mechanisms of your findings.
- Emphasize the new and important aspects of your study and put your findings in the context of the totality of the relevant evidence.
- State the limitations of your study, and explore the implications of your findings for future research and for clinical practice or policy.
- Do not repeat data or other information given in other parts of the manuscript in detail, such as in the Introduction or the Results section.
- Discuss the practical utility of the reported results.
- Discuss generalizability of your findings.
- Discuss the clinical utility of your findings ().
Author Response
Reviewer 1:
- The title is too long, and the essential information is lost.
Author response: Thank you for your critical comment. The title has changed to
“Therapeutic Decision Making in Prevascular Mediastinal Tumors Using CT Radiomics and Clinical Features: Upfront Surgery or Pretreatment Needle Biopsy?”
- Simple Summary is missing
Author response: Simple Summary was added as follows:
“Initial management approaches for prevascular mediastinal tumors (PMTs) can be divided into two categories: direct surgery and core needle biopsy (CNB). Although the gold standard diagnostic method is histopathological examination, the selection of the initial management between direct surgery and CNB is more urgent ant for patients with PMTs, compared with the definite diagnosis of PMT subtypes. The study aimed to develop clinical-radiomics machine learning (ML) classification models for differentiating patients who needed direct surgery from patients who needed CNB among patients with PMTs. An ensemble learning model, combining 5 ML models, had a classification accuracy of 90.4% (95% CI = 87.9 to 93.0%), which significantly outperformed clinical diagnosis (86.1%), which may be used as clinical decision support system to facilitate the selection of initial management of PMT.”
- Abstract
- Write the aim at the past tense.
- Provide the 95% CI for accuracy.
- The methods are not clear.
- The results are not sufficiently specific.
Author response: Thank you for your comments. The entire Abstract was thoroughly revised as follows:
“Abstract: The study aimed to develop machine learning (ML) classification models for differentiating patients who needed direct surgery from patients who needed core needle biopsy among patients with prevascular mediastinal tumor (PMT). Patients with PMT and receiving contrast-enhanced computed tomography (CECT) scan and initial management for PMT between January 2010 and December 2020 were included in this retrospective study. Fourteen ML algorithms were used to construct candidate classification models via voting ensemble approach, based on preoperative clinical data and radiomic features extracted from CECT. The classification accuracy of clinical diagnosis was 86.1%. The first ensemble learning model was built by randomly choosing 7 ML models from a set of 14 ML models, and had a classification accuracy of 88.0% (95% CI = 85.8 to 90.3%). The second ensemble learning model was the combination of 5 ML models, including NeuralNetFastAI, NeuralNetTorch, RandomForest with Entropy, RandomForest with Gini, and XGBoost, and had a classification accuracy of 90.4% (95% CI = 87.9 to 93.0%), which significantly outperformed clinical diagnosis. Due to the superior performance, the voting ensemble learning clinical-radiomic classification model may be used as a clinical decision support system to facilitate the selection of initial management of PMT.”
Introduction
- "In clinical practice, clinicians predict PMT subtypes based on preoperative clinical and radio-logic indicators and make corresponding recommendations for the initial management" please clarify the process.
Author response: Thank you for your critical comment. The corresponding statement has been revised for clarity as follows:
“Clinicians take age, preoperative clinical characteristics, and preoperative radiological features into consideration while making a clinical diagnosis. Thymic epithelial tumors are more common in adults aged 50-70 years; lymphoma and malignant germ cell tumors are more common in adults aged 20-40 years. Patients with malignant germ cell tumor often have abnormal serum levels of alpha-fetoprotein (AFP) and human chorionic gonadotropin (HCG). Regarding radiological features, benign teratoma often has a fat component, lymphoma often has multiple mediastinal lymphadenopathy, resectable thymoma often has a rounder shape with regular margin, and unresectable thymoma and thymic carcinoma often have great vascular encasement. Clinicians then recommend the suitable initial management based on the clinical diagnosis.”
- "misclassifications can occur" how often?
Author response: Thank you for your comment. The corresponding statement has been revised as follows:
“A misclassification rate of 13.9% was noted in the current retrospective study, which resulted in unnecessary CNB and surgical resection with a negative impact on patient health.”
- "extensively" did not fit with two references.
Author response: Thank you for pointing it out. “Extensively” was deleted.
- "combined predictive models using both clinical and radiological features have been constructed to improve the differential diagnostic performance" briefly present them, along with advantages and disadvantages. The need of a new method must be appropriately substantiated.
Author response: Thank you for your comment. The corresponding statements have been revised for clarity as follows:
“Liu et al. [26] reported that a radiomics-based ML model for differentiating anterior mediastinal cysts from thymomas has a sensitivity of 89.3%, while the sensitivity of ML model can be improved to 92.3% if both clinical and radiological features were used to construct ML model. Recently, several clinical-radiomics models have been developed to predict pathological subtypes of PMTs [27], to differentiate low-risk from high-risk thymomas [28], and to distingue TETs from other PMT subtypes [29], thereby facilitating clinical diagnosis. But, the gold standard diagnostic method is histopathological examination. In clinical practice, the selection of the initial management between direct surgery and CNB is a more urgent issue needed to be solved quickly, compared to the definite diagnosis of PMT subtypes. Imaging findings and clinical characteristics may be useful to differentiate surgical cases from nonsurgical cases [30]. Therefore, this retrospective study aimed to develop ML classification models using preoperative clinical features and radiomics to differentiate Category 1 from Category 2 patients. These expected classification models will contribute to the development of an ML-based clinical decision support system to facilitate the selection of the appropriate initial management for each individual patient with PMT.”
- Write the aim of the study at the past tense.
Author response: Thank you for pointing it out. It was corrected.
Methods
- "However, patients without complete contrast-enhanced computed tomography (CECT)". What about incomplete "pre-operative clinical features" data?
Author response: Thank you for your critical comment. Among clinical data, only serum tumor markers had missing data, because tumor markers are not part of routine blood tests. Abnormal serum levels of AFP and HCG are highly associated with malignant germ cell tumor. Clinicians may order tumor marker test only if patient is suspected to have malignant germ cell tumor. In this retrospective study, serum tumor marker levels were categorized as normal, abnormal, or missing, because patients with malignant germ cell tumor were included.
The above information was added to the subsection of 2.2. Clinical diagnosis and clinical data collection as follows:
“Serum tumor marker levels were categorized as normal, abnormal, or missing. It is worth noting that tumor markers are not part of routine blood tests, and clinicians may order them for patients with suspected malignant germ cell tumor.”
- "Clinical diagnosis and the subsequent treatment selection were based on baseline clinical data and preoperative CECT." the gold standard diagnostic in medicine is histopathological examination. Without such an exam the accuracy of any model is questionable.
Author response: Thank you for your critical comment. We agree that “the gold standard diagnostic in medicine is histopathological examination.” In this retrospective study, the performance of ML models was assessed based on histopathological examination. The gold standard is articulated in the Introduction section as follows:
“But, the gold standard diagnostic method is histopathological examination. In clinical practice, the selection of the initial management between direct surgery and CNB is a more urgent issue needed to be solved quickly, compared to the definite diagnosis of PMT subtypes.”
As mentioned in the subsection of 2.1. Patient selection and classification, all included patients had pathologically confirmed PMT
“Eligible patients were divided into two categories based on the final pathology reports. Category 1 patients had pathologically confirmed resectable TET, thymic hyperplasia, cyst, lymphangioma, thymolipoma, or teratoma that was to undergo direct surgery. Category 2 patients had pathologically confirmed unresectable TET, lymphoma, Castleman disease, or malignant germ cell tumor and were to receive CNB first.”
In addition, in the subsection of 3.1. Patient selection and grouping, the importance of pathological exmaination was articulated.
“A total of 375 eligible patients were included in the study. According to the final pathology reports, 182 patients were to undergo direct surgery (Category 1), and 193 patient were to undergo CNB first (Category 2).”
The result of clinical diagnosis was compared to histopathological examination (gold standard) to raise the need of ML classification models for the selection of the suitable initial management. This information was stated in the subsection of 3.3. Classification accuracy for clinical diagnosis.
“With reference to the final pathology reports (gold standard), the classification accuracy of the clinical diagnosis was 86.1% (323/375).”
- "It's" should be read as "It is"
Author response: Thank you for pointing it out. It was corrected.
- "Intravenous contrast was administered (injection dose 60-120 mL at a rate of 1.5 ml/s) was performed" please rephrase for clarity.
Author response: Thank you for your critical comment. The statement was revised as follows:
“Contrast medium (60 to 120 ml) was intravenously administered at a rate of 1.5ml/s, followed by a 20 mL saline flush.”
- "3D radiomic" do not start a sentence with an abbreviation.
Author response: Thank you for pointing it out. It was corrected as follows:
“The open-source platform PyRadiomics was used to extract 3D radiomic features from the segmentation of the PMTs on the images [33].”
- how radiomic features were selected?
Author response: The least absolute shrinkage and selection operator (LASSO) regression analysis is a feature selection method based on a linear regression model, and has been used in the development of clinical-radiomics ML models. This information as added to the subsection of 2.4. Radiomic feature extraction and LASSO regression as follows:
“The least absolute shrinkage and selection operator (LASSO) regression analysis is a feature selection method based on a linear regression model. Both the 851 radiomic features and all the above clinical data were entered into the LASSO regression for variable selection, as previously described [28, 29].”
- How the algorithms were selected?
Author response: Those 14 algorithms are built-in algorithms of the AutoGluon Tabular classifier. The corresponding statement was revised for clarity in the subsection of 2.5. ML model building as follows:
“Python version 3.8.9 with AutoGluon Tubular classifier version 0.8.2 [34], was used to model a variety of built-in ML algorithms…”
- "Continuous variables are presented as mean ± standard deviation" regardless the distribution?
Author response: Thank you for your comment. The subsection of 2.6 Statistical analysis was revised for clarity as follows:
“Age is expressed as median (Q1, Q3), and between-group difference was examined using Mann-Whitney U test. Categorical variables are expressed as counts (percentages), and between-group differences were compared using Chi-square and Fisher's exact test.”
In addition, Table 1 was updated accordingly.
- It is unclear if you have or not a training and a validation set.
Author response: Thank you for your comment. This study did not have a training dataset and a distinct validation dataset. Instead, 10-fold cross-validation was utilized in this study. The dataset was divided into 10 subsets. Each subset took turns to be validation set. This information was added to the subsection of 2.5. ML model building as follows:
“First, all clinical variables and radiomic features were used to train and validate 14 different ML models using 10-fold cross validation, a common approach for evaluating the performance of ML prediction models [35-37]. The dataset was divided into 10 subsets. The model was trained on 9 randomly selected subsets, and was validated on the remaining subset. The model was then trained and validated in the same way 10 times. Thus, the macro F1 score, macro precision, macro recall, accuracy, and area under the receiver operating characteristic curve (AUROC) calculated from each time were averaged to estimate the generalization performance of the model, as previously described [38].”
In addition, the lack of a training and a validation datasets was recognized as a limitation in the Discussion section as follows:
“In addition, this retrospective study did not have a training dataset and a separated validation dataset because of a spectrum of many PMT subtypes. Instead, 10-fold cross-validation was utilized to train and validate ML models.”
Results
- In the flow something is wrong ... 408 eligible ... 375 without contrast CT scan and 375 included in the study.
Author response: Thank you for pointing out the typo. The Figure 2 was corrected.
- A result must stay either in text or in table/figure (see Figure 3).
Author response: Thank you for your comment. Figure 3 was deleted.
- "In the present study, three different approaches were used to develop ML classifica-tion models to discriminate between Category 1 patients and Category 2 patients. First, all clinical variables and radiomic features were used to train 14 different ML models us-ing 10-fold cross validation." duplicate information that belngs to the Methods section; please delete.
Author response: Thank you for your comment. The corresponding information was relocated to the subsection of 2.5. ML model building, and has been revised for clarity and conciseness as follows:
“Three different approaches were used to develop ML classification models to discriminate between Category 1 patients and Category 2 patients. First, all clinical variables and radiomic features were used to train and validate 14 different ML models using 10-fold cross validation, a common approach for evaluating the performance of ML prediction models [35-37]. The dataset was divided into 10 subsets. The model was trained on 9 randomly selected subsets, and was validated on the remaining subset. The model was then trained and validated in the same way 10 times. Thus, the macro F1 score, macro precision, macro recall, accuracy, and area under the receiver operating characteristic curve (AUROC) calculated from each time were averaged to estimate the generalization performance of the model, as previously described [38].”
- "All clinical variables and radiomic features were then subjected to LASSO regression 20 times, resulting in 20 different combinations of selected variables, designated as Selec-tion_1 to Selection_20. The features selected by each run of LASSO selection, as well as allfeatures without LASSO selection, were independently used to build the CatBoost model, in order to further improve the classification performance of the CatBoost model." this information belongs to the Methods section.
Author response: Thank you for your comment. The above information was relocated to the subsection of 2.5. ML model building, and has been revised as follows:
“Additionally, LASSO regression was applied to all clinical variables and radiomic features, resulting in 20 combinations of selected variables based on various lambda values. These combinations are labeled as Selection_1 to Selection_20. The features selected by LASSO selection, as well as all features without LASSO selection, were independently used to build the CatBoost model.”
- "Voting ensemble learning models were also built. The features selected by Selec-tion_3 and all features without LASSO selection were used independently to build voting ensemble learning models. The classification performances of all combinations of 3, 5, 7, or 9 ML models were then separately averaged." Methods
Author response: Thank you for your comment. The above information was relocated to the subsection of 2.5. ML model building, and has been revised as follows:
“Subsequently, we constructed voting ensemble ML models based using Python version 3.8.9. The final decision was made by majority voting. The features selected by Selection_3 and all features without LASSO selection were used independently to build voting ensemble learning models. The performance of all combinations of 3, 5, 7, or 9 ML models, which were randomly selected from a total of 14 ML models, was evaluated. Two voting ensemble approaches were used to select the optimal ML models. First, the mean classification performances of these combinations of 3, 5, 7, or 9 were averaged separately. The combination of a given number of ML models with the highest mean accuracy was identified as the final model. Second, the single combined ML model with the highest accuracy was identified as the optimal voting ensemble learning model. The construction workflow of the voting ensemble learning models is shown in Figure 1.”
- The Results section is a mixture of Methods and Results, sometimes with results duplicated in text and tables/images.
Author response: Thank you for your valuable comments. The Methods and Results sections have been revised accordingly.
Discussion
- Do not write again the results.
Author response: Thank you for your critical comment. The entire Discussion section has been revised accordingly.
- "We used 10-fold cross-validation, a common approach for evaluating the perfor-mance of ML prediction models [44-46]. The dataset was divided into 10 subsets, and the model was trained and validated 10 times. Each time, the model was trained on 9 randomly selected subsets, and was validated on the remaining subset. Thus, the macro F1 score, macro precision, macro recall, accuracy, and AUROC calculated from each time were averaged to estimate the generalization performance of the model." this information belongs to Methods section.
Author response: Thank you for your valuable comments. The information was relocated to the subsection of 2.5. ML model building and has been revised as follows:
“First, all clinical variables and radiomic features were used to train and validate 14 different ML models using 10-fold cross validation, a common approach for evaluating the performance of ML prediction models [35-37]. The dataset was divided into 10 subsets. The model was trained on 9 randomly selected subsets, and was validated on the remaining subset. The model was then trained and validated in the same way 10 times. Thus, the macro F1 score, macro precision, macro recall, accuracy, and area under the receiver operating characteristic curve (AUROC) calculated from each time were averaged to estimate the generalization performance of the model, as previously described [38].”
- Begin the discussion by briefly summarizing the main findings.
Author response: Thank you for your essential comment. The main findings were summarized as follows:
“This retrospective study revealed that clinical diagnosis had an accuracy of 86.1%. Of 14 ML algorithms evaluated, the CatBoost model achieved the highest accuracy of 84.2%. The first voting ensemble ML model, which randomly selected 7 ML models from the set of 14, had an average classification accuracy of 88.0%. The second voting ensemble ML model, a specific combination of 5 ML models, achieved an accuracy of 90.4%. Particularly, the classification accuracy of the second voting ensemble ML model was significantly higher than that of clinical diagnosis.”
- Explore possible mechanisms of your findings.
Author response: Thank you for your insightful comment.
- The clinical significance of radiomic features was briefly discussed as follows:
“Automated extraction of a huge amount of quantitative radiomic features allows analyzing subtle characteristics of tumors on images, such as shpericity, diameter, homogeneity, and calcification, thereby facilitating disease differential diagnosis [46]. Resectable TETs usually have regular and smooth borders and smaller diameters, while lymphomas, malignant germ cell tumors, and unresectable TETs tend to have larger diameters and irregular borders. Lymphoma is more homogenous, but TET and teratoma are more heterogeneous and may be calcified, resulting in difference in the Hounsfield Unit value.”
- The clinical significance of clinical features was briefly discussed as follows:
“Notably, only one clinical feature, the serum level of LDH, was extracted by LASSO in Selection_3. Although patients with malignant germ cell tumor often have abnormal levels of AFP and HCG, those clinical features with high specificity were not selected by LASSO, which may in part due to few patients with malignant germ cell tumor in the study population. Although LDH has a low specificity for differential diagnosis of PMT subtype, PMTs with higher malignancy and fast growth, such as lymphoma, high-grade thymoma, thymic carcinoma, and malignant germ cell tumor, usually have abnormal LDH levels. Therefore, serum LDH may be considered for inclusion in routine preoperative blood tests. In support of this suggestion, an elevated serum LDH level has been reported in conditions such as mediastinal seminoma, mediastinal lymphoma, and leukocytosis [47].”
- Emphasize the new and important aspects of your study and put your findings in the context of the totality of the relevant evidence.
Author response: Thank you for your critical comments.
- Currently, most clinical-radiomics ML models have been developed to predict PMT subtypes, thereby facilitating clinical diagnosis. But, the gold standard diagnostic method is histopathological examination. In clinical practice, the selection of the initial management between direct surgery and CNB is a more urgent issue needed to be solved quickly, compared to the definite diagnosis of PMT subtypes. Thus, this study aimed to develop clinical-radiomics ML model to differentiate Category 1 patients who are recommended to undergo immediate surgical resection from Category 2 patients who are advised to take CNB first. The novelty of this study was explained in the Introduction section as follows:
“Recently, several clinical-radiomics models have been developed to predict pathological subtypes of PMTs [27], to differentiate low-risk from high-risk thymomas [28], and to distingue TEEs from other PMT subtypes [29], thereby facilitating clinical diagnosis. But, the gold standard diagnostic method is histopathological examination. In clinical practice, the selection of the initial management between direct surgery and CNB is a more urgent issue needed to be solved quickly, compared to the definite diagnosis of PMT subtypes. Imaging findings and clinical characteristics may be useful to differentiate surgical cases from nonsurgical cases [30]. Therefore, this retrospective study aimed to develop ML classification models using preoperative clinical features and radiomics to differentiate Category 1 from Category 2 patients. These expected classification models will contribute to the development of an ML-based clinical decision support system to facilitate the selection of the appropriate initial management for each individual patient with PMT.”
- Some examples of ensemble learning models were discussed in the Discussion section as follows:
“ML algorithms used in this study have been commonly utilized to develop classification models for various diseases [48-52]. Ensemble learning techniques have been widely used in clinical practice to combine several ML models to create a stronger model with superior performance. The ensemble model has a better predictive accuracy than those of individual models [49]. RandomForest, CatBoost and XGBoost were used to develop an ensemble model for predicting pre-cancer in pre- and post-menopausal women, with an accuracy of 94% [50]. In particular, voting ensemble methods have been used to improve the performance of prediction models for detection of cervical cancer and brain tumor [53,54], prognosis of non-small cell lung cancer [52], and diagnosis and prognosis of acute coronary syndrome [55]. In our research, we applied voting ensemble methods to construct an ensemble learning models with classification accuracy of 90.4%, which significantly outperformed clinical diagnosis.”
- State the limitations of your study, and explore the implications of your findings for future research and for clinical practice or policy.
Author response: Thank you for your critical comment. The limitations and future research directions were discussed as follows:
“Some limitations need to be addressed. First of all, the findings of this single-institution retrospective study need to be confirmed by other studies conducted in distinct geographic areas. In addition, this retrospective study did not have a training dataset and a separated validation dataset because of a spectrum of many PMT subtypes. Instead, 10-fold cross-validation was utilized to train and validate ML models. The mean classification failure rates of ensemble learning models could be calculated in the present retrospective study; however, patients at high risk of misclassification and underlying reasons could not be investigated. Moreover, due to the etiologic diversity of PMTs [6,56], many PMT subtypes are extremely rare and may not be considered in the binary classification system for initial treatment. Therefore, large-scale multicenter studies are warranted to evaluate the feasibility of ensemble ML classification model as a clinical decision support system to facilitate the selection of initial treatment for patients with PMTs.
Furthermore, some unresectable TET cases may receive neoadjuvant chemotherapy first. Once the tumor size is reduced, surgical resection will be performed to improve proportion of R0 resection and the prognosis. However, due to the small number of patients undergoing neoadjuvant chemotherapy followed by surgery, large-scale multicenter studies are needed to develop a reliable model for predicting patients with TET who may undergo neoadjuvant chemotherapy followed by surgical resection.”
- Do not repeat data or other information given in other parts of the manuscript in detail, such as in the Introduction or the Results section.
Author response: Thank you for your critical comment. The entire Discussion section has been revised accordingly.
- Discuss the practical utility of the reported results.
Author response: Thank you for your comment. The potential practical utility of the findings was discussed as follows:
“Compared with other clinical-radiomics model studies that limited specific PMT histology types [27-29], this study did not set restrictions on PMT subtypes, including12 different PMT subtypes. In addition, unlike other clinical-radiomics model studies aimed to improve differential diagnosis [27-29], the developed voting ensemble ML model may be utilized as a clinical decision support system to help the selection of initial management for patients with PMTs.
- Discuss generalizability of your findings.
Author response: Thank you for your essential comment. Generalizability was discussed as follows:
“First of all, the findings of this single-institution retrospective study need to be confirmed by other studies conducted in distinct geographic areas.”
- Discuss the clinical utility of your findings (https://doi.org/10.1155/2019/1891569).
Author response: Appreciate your valuable comment. The potential clinical utility of this study is to facilitate the development of clinical decision support systems, which was discussed as follows:
“Although the gold standard diagnostic method is histopathological examination, the initial management for patients with PMT is determined based on clinical diagnosis, but not pathological diagnosis, indicating the importance of clinical diagnostic tests [39]. In clinical practice, a multidisciplinary team works together to reach a consensus on the clinical diagnosis and initial management for patients with PMT. This consensus-based decision-making process is time- and labor-consuming. Moreover, discrepancy in clinical diagnostic suggestions might occur in patients aged 30-40 or older than 70, who have normal levels of tumor markers but no specific radiological features. ML classification models with superior classification accuracy may serve as clinical decision support systems.”
Reviewer 2 Report
Comments and Suggestions for Authors
1. The criteria for patient inclusion and exclusion need more explicit justification. For example, why was the age threshold set at 20 years? How representative is this sample of the broader population with PMT?
2. The involvement of over 20 clinicians in clinical diagnosis might introduce variability. How was the consistency of clinical diagnosis ensured among these clinicians? Additionally, how were missing data and the non-routine nature of tumor marker tests handled in the analysis?
3. The use of four different CT scanners may introduce variations in image quality and characteristics. How were these variations accounted for in the analysis, and what impact might they have on the robustness of the radiomic features?
4. The extraction of 851 radiomic features is extensive, and the relevance of all these features to the study needs to be discussed. Moreover, the choice of the LASSO regression technique should be justified. Were other feature selection methods considered, and how sensitive are the results to the choice of regularization?
5. The use of multiple ML algorithms is appropriate, but the rationale behind the selection of specific algorithms needs clarification. How were hyperparameters optimized, and what measures were taken to prevent overfitting or underfitting? The performance metrics used for model evaluation should also be justified.
6. The decision to use a voting ensemble model is sound, but the process of selecting the optimal combination of ML models for the ensemble should be detailed further. Why were specific numbers of models (3, 5, 7, or 9) chosen, and what is the impact of this choice on the final results?
7. More references on radiomics-based analysis studies should be added to attract a broader readership i.e., PMID: 36717518, PMID: 34771562.
8. The statistical analyses section is brief, and more details about the statistical tests performed and the rationale behind choosing specific tests are needed. Additionally, the potential limitations and assumptions of the statistical models used should be discussed.
9. The study needs to address how well the findings can be generalized to other populations or settings. Were there any demographic or clinical characteristics in the study population that might limit the generalizability of the results?
10. The study should discuss the potential clinical implications of the findings. How could the proposed ML models impact decision-making in the clinical setting, and what further validation or clinical studies are needed before implementation?
11. The authors should compare the predictive performance to previously published works on the same problem/data.
12. When comparing the performance among methods/models, the authors should perform some statistical tests to see significant differences.
Comments on the Quality of English LanguageEnglish writing should be improved.
Author Response
- The criteria for patient inclusion and exclusion need more explicit justification. For example, why was the age threshold set at 20 years? How representative is this sample of the broader population with PMT?
Author response: Thank you for your comments.
- Patients aged under 20 years are likely to have lymphoma, and are usually recommended to receive core needle biopsy (CNB) first.
- Based on the final pathology reports, this study included a broad spectrum of PMT subtypes, including resectable thymoma, resectable thymic carcinoma, thymic hyperplasia, cyst, teratoma, thymolipoma, lymphangioma, unresectable thymoma, unresectable thymic carcinoma, lymphoma, malignant germ cell tumor, and Castleman disease (Table 1).
- The involvement of over 20 clinicians in clinical diagnosis might introduce variability. How was the consistency of clinical diagnosis ensured among these clinicians? Additionally, how were missing data and the non-routine nature of tumor marker tests handled in the analysis?
Author response: Thank you for your critical comments.
- We were not able to assess the consistency in clinical diagnostic suggestions among clinicians due to the retrospective nature of this study. Although clinicians usually reach consensus in most cases, discrepancy in clinical diagnostic suggestions often occur in patients aged 30-40 or older than 70, who have normal levels of tumor markers but no specific radiological features. Thus, this study aimed to develop a ML-based clinical decision support system to facilitate the selection of the most appropriate initial treatment for patients with PMT.
The above information was added to the second paragraph in the Discussion section as follows:
“In clinical practice, a multidisciplinary team works together to reach a consensus on the clinical diagnosis and initial management for patients with suspected PMT. This consensus-based decision-making process is time- and labor-consuming. Moreover, discrepancy in clinical diagnostic suggestions often occur in patients aged 30-40 or older than 70, who have normal levels of tumor markers but no specific radiological features. ML classification models with superior classification accuracy may serve as clinical decision support systems.”
- Among clinical data, only serum tumor markers had missing data, because tumor markers are not part of routine blood tests. Abnormal serum levels of AFP and HCG are highly associated with malignant germ cell tumor. Clinicians may order tumor marker test only if patient is suspected to have malignant germ cell tumor. In this retrospective study, serum tumor marker levels were categorized as normal, abnormal, or missing, because patients with malignant germ cell tumor were included.
The above information was integrated into the subsection of 2.2. Clinical diagnosis and clinical data collection as follows:
“Serum tumor marker levels were categorized as normal, abnormal, or missing. It is worth noting that tumor markers are not part of routine blood tests, and clinicians may order them for patients with suspected malignant germ cell tumor.”
- The use of four different CT scanners may introduce variations in image quality and characteristics. How were these variations accounted for in the analysis, and what impact might they have on the robustness of the radiomic features?
Author response: Thank you for your critical comment. PMTs are relatively rare and heterogeneous tumors. In this single-center retrospective study, we were not able to evaluate the influence of different CT scanners in image quality and characteristics in patients with the same PMT subtype. However, the use of different CT scanners does not affect performance of ML models for detecting lung nodule, but reconstruction kernel affects radiomic feature selection and model performance.
The above information was integrated into the Discussion section as follows:
“In this retrospective study, CECT images were generated by four different CT scanners. Because PMTs are rare and heterogeneous tumors, we were not able to evaluate the influence of different CT scanners in image quality and characteristics in patients with the same PMT subtype. Notably, the use of different CT scanners does not affect performance of ML models for detecting lung nodule [45], but reconstruction kernel affects radiomic feature selection and model performance [31]. Thus, in this study, all CECT raw data were reconstructed with a smooth standard convolution kernel (B40f) to reduce variations in image characteristics.”
In this retrospective study, all CECT raw data were reconstructed with a smooth standard convolution kernel (B40f) to reduce variations in image quality and characteristics. This information was added to the subsection of 2.3. CECT Image Acquisition and Preprocessing as follows:
“All images were reconstructed in 5 mm slices with a smooth standard convolution kernel (B40f), as previously described [31].”
- The extraction of 851 radiomic features is extensive, and the relevance of all these features to the study needs to be discussed. Moreover, the choice of the LASSO regression technique should be justified. Were other feature selection methods considered, and how sensitive are the results to the choice of regularization?
Author response: Thank you for your critical comments.
- The clinical significance of radiomic features was briefly discussed as follows:
“Automated extraction of a huge amount of quantitative radiomic features allows analyzing subtle characteristics of tumors on images, such as shpericity, diameter, homogeneity, and calcification, thereby facilitating disease differential diagnosis [46]. Resectable TETs usually have regular and smooth borders and smaller diameters, while lymphomas, malignant germ cell tumors, and unresectable TETs tend to have larger diameters and irregular borders. Lymphoma is more homogenous, but TET and teratoma are more heterogeneous and may be calcified, resulting in difference in the Hounsfield Unit value.”
- No other feature selection method was utilized. The use of the LAASO regression was justified in the subsection of 4. Radiomic feature extraction and LASSO regression as follows:
“The least absolute shrinkage and selection operator (LASSO) regression analysis is a feature selection method based on a linear regression model. Both the 851 radiomic features and all the above clinical data were entered into the LASSO regression for variable selection, as previously described [28, 29].”
- In the LASSO regression analysis, as lambda increases from left to right, coefficients often shrink to 0. The variables that do not shrink to 0 represent important variables, and are selected by LASSO regression. As shown in the figure below, log lambda = Ln (0.0254) = -3.675, indicating that when log lambda = -3.675, important variables are selected by the LASSO regression.
Due to the small sample size, 10-fold cross validation was used to train and validate ML models in this study. The standard deviations (SDs) of almost all performance outcomes of ML models were within 5%, suggesting stability.
- Table 3 was amended by adding the lambda value of each selection. The Selection_3 had a lambda of 0.025354. Ln (0.025354) = -3.675. Thus, the Selection_3 was the optimal selection. This information was added into the subsection of 4. The individual ML model as follows:
“The CatBoost model with the third LASSO selection (Selection_3) had the best classification performance, with a lambda of 0.025354, a Ln (lambda) of -3.675, a macro F1 score of 0.8417 ± 0.0426, an accuracy of 0.8422 ± 0.0423, and an AUROC of 0.9114 ± 0.0324 (Table 3).”
- The use of multiple ML algorithms is appropriate, but the rationale behind the selection of specific algorithms needs clarification. How were hyperparameters optimized, and what measures were taken to prevent overfitting or underfitting? The performance metrics used for model evaluation should also be justified.
Author response: Thank you for your critical comments.
- Those 14 algorithms are built-in algorithms of the AutoGluon Tabular classifier. The corresponding statement was revised for clarity in the subsection of 5. ML model building as follows:
“Python version 3.8.9 with AutoGluon Tubular classifier version 0.8.2 [34], was used to model a variety of built-in ML algorithms, including CatBoost, ExtraTrees with Entropy, ExtraTrees with Gini, Kneighbors with Distance Weights, Kneighbors with Uniform Weights, LightGBM, LightGBMLarge, LightGBM with ExtraTress, NeurlNetFastAI, NeuralNetTorch, RandomForest with Entropy, RandomForest with Gini, WeightedEnsemble_L2, and XGBoost to develop classification models.”
- The LASSO regression analysis was clarified in the subsection of 4. Radiomic feature extraction and LASSO regression as follows:
“The variables with non-zero coefficients and optimal lambda values were selected by the LASSO regression to build ML classification models to distinguish Category 1 patients from Category 2 patients.
- The performance metrics used for model evaluation was justified in the subsection of 5. ML model building as follows:
“Three different approaches were used to develop ML classification models to discriminate between Category 1 patients and Category 2 patients. First, all clinical variables and radiomic features were used to train and validate 14 different ML models using 10-fold cross validation, a common approach for evaluating the performance of ML prediction models [35-37]. The dataset was divided into 10 subsets. The model was trained on 9 randomly selected subsets, and was validated on the remaining subset. The model was then trained and validated in the same way 10 times. Thus, the macro F1 score, macro precision, macro recall, accuracy, and area under the receiver operating characteristic curve (AUROC) calculated from each time were averaged to estimate the generalization performance of the model, as previously described [38].”
- The decision to use a voting ensemble model is sound, but the process of selecting the optimal combination of ML models for the ensemble should be detailed further. Why were specific numbers of models (3, 5, 7, or 9) chosen, and what is the impact of this choice on the final results?
Author response: Thank you for your essential comments.
- The construction of voting ensemble ML models was detailed as follows:
“Subsequently, we constructed voting ensemble ML models based using Python version 3.8.9. The final decision was made by majority voting. The features selected by Selection_3 and all features without LASSO selection were used independently to build voting ensemble learning models. The performance of all combinations of 3, 5, 7, or 9 ML models, which were randomly selected from a total of 14 ML models, was evaluated. Two voting ensemble approaches were used to select the optimal ML models. First, the mean classification performances of these combinations of 3, 5, 7, or 9 were averaged separately. The combination of a given number of ML models with the highest mean accuracy was identified as the final model. Second, the single combined ML model with the highest accuracy was identified as the optimal voting ensemble learning model. The construction workflow of the voting ensemble learning models is shown in Figure 1.”
- If an even number of models is selected for major voting, the number of votes may be equal and the decision cannot be made. Therefore, we only chose an odd number (3, 5, 7, 9) for voting. We did not choose 11 and 13 because of the huge amount of calculation.
- More references on radiomics-based analysis studies should be added to attract a broader readership i.e., PMID: 36717518, PMID: 34771562.
Author response: Thank you for your valuable comment. The following information was added to the Discussion section.
“Therefore, ML classification models with superior classification accuracy may serve as clinical decision support systems for differential diagnosis and management. Notably, radiomics-based ML models have been developed to predict overall survival in various cancer sites [40], and to predict management in lower-grade gliomas [41].”
- The statistical analyses section is brief, and more details about the statistical tests performed and the rationale behind choosing specific tests are needed. Additionally, the potential limitations and assumptions of the statistical models used should be discussed.
Author response: Thank you for your valuable comments.
- The subsection of 6 Statistical analysis was revised for clarity as follows:
“Age is expressed as median (Q1, Q3), and between-group difference was examined using Mann-Whitney U test. Categorical variables are expressed as counts (percentages), and between-group differences were compared using Chi-square and Fisher's exact test. As 10-fold cross-validation was used in this study, macro F1 score, macro precision, macro recall, accuracy, and AUROC were expressed as mean ± SD. Statistical significance was set at a p-value of 0.05 or 95% confidence interval (95% CI). Statistical analysis of ML models was performed using statsmodels version 0.13.1.”
The process of 10-fold cross-validation was explained in the subsection of 2.5. ML model building as follows:
“First, all clinical variables and radiomic features were used to train and validate 14 different ML models using 10-fold cross validation, a common approach for evaluating the performance of ML prediction models [35-37]. The dataset was divided into 10 subsets. The model was trained on 9 randomly selected subsets, and was validated on the remaining subset. The model was then trained and validated in the same way 10 times. Thus, the macro F1 score, macro precision, macro recall, accuracy, and area under the receiver operating characteristic curve (AUROC) calculated from each time were averaged to estimate the generalization performance of the model, as previously described [38].”
- The use of 10-fold cross-validation was recognized as a limitation of this study as follows:
“In addition, this retrospective study did not have a training dataset and a separated validation dataset because of a spectrum of many PMT subtypes. Instead, 10-fold cross-validation was utilized to train and validate ML models. The mean classification failure rates of ensemble learning models could be calculated in the present retrospective study; however, patients at high risk of misclassification and underlying reasons could not be investigated. Moreover, due to the etiologic diversity of PMTs [6,56], many PMT subtypes are extremely rare and may not be considered in the binary classification system for initial treatment. Therefore, large-scale multicenter studies are warranted to evaluate the feasibility of ensemble ML classification model as a clinical decision support system to facilitate the selection of initial treatment for patients with PMTs.”
- The study needs to address how well the findings can be generalized to other populations or settings. Were there any demographic or clinical characteristics in the study population that might limit the generalizability of the results?
Author response: Thank you for your comment. Generalizability was discussed as follows:
“First of all, the findings of this single-institution retrospective study need to be confirmed by other studies conducted in distinct geographic areas.”
- The study should discuss the potential clinical implications of the findings. How could the proposed ML models impact decision-making in the clinical setting, and what further validation or clinical studies are needed before implementation?
Author response: Thank you for your valuable comments.
- The potential clinical application of the findings was discussed as follows:
“Compared with other clinical-radiomics model studies that limited specific PMT histology types [27-29], this study did not set restrictions on PMT subtypes, including12 different PMT subtypes. In addition, unlike other clinical-radiomics model studies aimed to improve differential diagnosis [27-29], the developed voting ensemble ML model may be utilized as a clinical decision support system to help the selection of initial management for patients with PMTs.”
- The future research directions were discussed as follows:
“Some limitations need to be addressed. First of all, the findings of this single-institution retrospective study need to be confirmed by other studies conducted in distinct geographic areas. In addition, this retrospective study did not have a training dataset and a separated validation dataset because of a spectrum of many PMT subtypes. Instead, 10-fold cross-validation was utilized to train and validate ML models. The mean classification failure rates of ensemble learning models could be calculated in the present retrospective study; however, patients at high risk of misclassification and underlying reasons could not be investigated. Moreover, due to the etiologic diversity of PMTs [6,56], many PMT subtypes are extremely rare and may not be considered in the binary classification system for initial treatment. Therefore, large-scale multicenter studies are warranted to evaluate the feasibility of ensemble ML classification model as a clinical decision support system to facilitate the selection of initial treatment for patients with PMTs.
Furthermore, some unresectable TET cases may receive neoadjuvant chemotherapy first. Once the tumor size is reduced, surgical resection will be performed to improve proportion of R0 resection and the prognosis. However, due to the small number of patients undergoing neoadjuvant chemotherapy followed by surgery, large-scale multicenter studies are needed to develop a reliable model for predicting patients with TET who may undergo neoadjuvant chemotherapy followed by surgical resection.”
- The authors should compare the predictive performance to previously published works on the same problem/data.
Author response: Thank you for your comment. Some examples of ensemble learning models were discussed as follows:
“ML algorithms used in this study have been commonly utilized to develop classification models for various diseases [48-52]. Ensemble learning techniques have been widely used in clinical practice to combine several ML models to create a stronger model with superior performance. The ensemble model has a better predictive accuracy than those of individual models [49]. RandomForest, CatBoost and XGBoost were used to develop an ensemble model for predicting pre-cancer in pre- and post-menopausal women, with an accuracy of 94% [50]. In particular, voting ensemble methods have been used to improve the performance of prediction models for detection of cervical cancer and brain tumor [53,54], prognosis of non-small cell lung cancer [52], and diagnosis and prognosis of acute coronary syndrome [55]. In our research, we applied voting ensemble methods to construct an ensemble learning models with classification accuracy of 90.4%, which significantly outperformed clinical diagnosis.”
- When comparing the performance among methods/models, the authors should perform some statistical tests to see significant differences.
Author response: Thank you for your important comment. The subsection of 2.6 Statistical analyses was revised for clarity as follows:
“Age is expressed as median (Q1, Q3), and between-group difference was examined using Mann-Whitney U test. Categorical variables are expressed as counts (percentages), and between-group differences were compared using Chi-square and Fisher's exact test. As 10-fold cross-validation was used in this study, macro F1 score, macro precision, macro recall, accuracy, and AUROC were expressed as mean ± SD. Statistical significance was set at a p-value of 0.05 or 95% confidence interval (95% CI). Statistical analysis of ML models was performed using statsmodels version 0.13.1.”
The corresponding statements in the Results section were revised for clarity as follows:
“However, the accuracy of the CatBoost model was 82.27 (95% CI = 79.6 to 84.9%) was significantly lower than that of clinical diagnosis (86.1%), because 86.1% did not fall within the 95% CI of the CatBoost model.”
“Nevertheless, the accuracy of the CatBoost model with Selection_3 was 84.2% (95% CI = 81.6 to 86.8%), which was still significantly lower than that of clinical diagnosis (86.1%).”
“The accuracy of this combining ML was 88.0% (95% CI = 85.8 to 90.3%) was not significant different from that of clinical diagnosis (86.1%)”.
“The accuracy of this ensemble learning classification model was 90.4% (95% CI = 87.9 to 93.0%) was significantly higher than that of clinical diagnosis (86.1%).”
- English writing should be improved.
Author response: Thank you for your important comment. The entire manuscript has been revised to improve the grammar and readability.
Reviewer 3 Report
Comments and Suggestions for Authors
· The merit of the proposed approach is supported by the results, but I miss on the paper a bit more discussion on why these techniques were chosen for this problem and had not been considered before. This however is more of a nitpicking than a detrimental comment.
· In the introduction, what key theoretical perspectives and empirical findings in the main literature have already informed the problem formulation? What major, unaddressed puzzle, controversy, or paradox does this research address?
· Why does it need to be addressed?
· Why it should be now - not in the past?
· Further, in the introduction, what is the recent knowledge gap of the main literature that the author needs to write this research? What we have known and what we have not known? What is missing from current works? Please explain and give examples!
· Why authors focused on the prevascular mediastinal tumor ?
· Below papers have some interesting implications that you could discuss in your introduction and how it relates to your work.
· Vulli, A.; et al.. Fine-Tuned DenseNet-169 for Breast Cancer Metastasis Prediction Using FastAI and 1-Cycle Policy. Sensors 2022, 22, 2988.
· GV Eswara Rao, B Rajitha, Parvathaneni Naga Srinivasu, Muhammad Fazal Ijaz, Marcin Woźniak. Hybrid framework for respiratory lung diseases detection based on classical CNN and quantum classifiers from chest X-rays. Biomedical Signal Processing and Control
· Praveen, S. Phani, et al. "ResNet-32 and FastAI for diagnoses of ductal carcinoma from 2D tissue slides." Scientific Reports 12.1 (2022): 20804.
· A Deep Transfer Learning Approach For Covid-19 Detection And Exploring A Sense Of Belonging With Diabetes. Ijaz Ahmad, Arcangelo Merla, Babar Shah, Ahmad Ali Alzubi, Mallak Ahmad AlZubi, Farman Ali. Frontiers in Public Health
· It would be interesting if the authors report the trade-off compared to other methods especially the computational complexity of the models. Some techniques require more memory space and take longer time, please elaborate on that.
· What are the practical implications of your research?
· What are the limitations of the present work?
· Conclusion is too short. Add more explanation.
Author Response
Reviewer 3:
- The merit of the proposed approach is supported by the results, but I miss on the paper a bit more discussion on why these techniques were chosen for this problem and had not been considered before. This however is more of a nitpicking than a detrimental comment.
Author response: Thank you for your critical comment.
- The 14 algorithms used in this study, are built-in algorithms of the AutoGluon Tabular classifier. The corresponding statement was revised for clarity in the subsection of 5. ML model building as follows:
“Python version 3.8.9 with AutoGluon Tubular classifier version 0.8.2 [34], was used to model a variety of built-in ML algorithms, including CatBoost, ExtraTrees with Entropy, ExtraTrees with Gini, Kneighbors with Distance Weights, Kneighbors with Uniform Weights, LightGBM, LightGBMLarge, LightGBM with ExtraTress, NeurlNetFastAI, NeuralNetTorch, RandomForest with Entropy, RandomForest with Gini, WeightedEnsemble_L2, and XGBoost to develop classification models.”
- The rationale of this research has been explained in detail in the Introduction section as follows:
“Given that PMTs are highly heterogeneous diseases, it's imperative to tailor different
management strategies to the specific subtypes [9,10]. For example, guidelines recommend medical treatment rather than surgical resection for lymphomas [11]. In contrast, resectable thymic epithelial tumors (TETs) require surgical resection to avoid the risk of tumor spread during a biopsy of an encapsulated thymoma [12]. Initial management approaches for PMTs can be divided into two categories: direct surgery and core needle biopsy (CNB). Category 1 includes patients with resectable TETs, cysts, thymic hyperplasia, teratomas, goiters, lymphangiomas, or thymolipomas, who are advised to undergo immediate surgical resection [13-15]. Category 2 includes patients with unresectable TETs, lymphomas, Castleman disease, or malignant germ cell tumors, who are recommended to start with CNB [15-17]. After a confirmed pathologic diagnosis, Category 2 patients may receive targeted therapy, chemotherapy, and/or radiotherapy [18,19].
Clinicians take age, preoperative clinical characteristics, and preoperative radiological features into consideration while making a clinical diagnosis. Thymic epithelial tumors are more common in adults aged 50-70 years; lymphoma and malignant germ cell tumors are more common in adults aged 20-40 years. Patients with malignant germ cell tumor often have abnormal serum levels of alpha-fetoprotein (AFP) and human chorionic gonadotropin (HCG). Regarding radiological features, benign teratoma often has a fat component, lymphoma often has multiple mediastinal lymphadenopathy, resectable thymoma often has a rounder shape with regular margin, and unresectable thymoma and thymic carcinoma often have great vascular encasement. Clinicians then recommend the suitable initial management based on the clinical diagnosis. The accuracy of such binary classification was confirmed by the final pathology report. A misclassification rate of 13.9% was noted in the current retrospective study, which resulted in unnecessary CNB and surgical resection with a negative impact on patient health.
With the advancement of machining learning (ML) technology, various image-based ML models have been developed to predict breast cancer metastasis [20], lung diseases [21,22], ductal carcinoma [23]. Furthermore, radiomics-based predictive models have been developed to differentiate between PMT subtypes [24,25]. Liu et al. [26] reported that a radiomics-based ML model for differentiating anterior mediastinal cysts from thymomas has a sensitivity of 89.3%, while the sensitivity of ML model can be improved to 92.3% if both clinical and radiological features were used to construct ML model. Recently, several clinical-radiomics models have been developed to predict pathological subtypes of PMTs [27], to differentiate low-risk from high-risk thymomas [28], and to distingue TETs from other PMT subtypes [29], thereby facilitating clinical diagnosis. But, the gold standard diagnostic method is histopathological examination. In clinical practice, the selection of the initial management between direct surgery and CNB is a more urgent issue needed to be solved quickly, compared to the definite diagnosis of PMT subtypes. Imaging findings and clinical characteristics may be useful to differentiate surgical cases from nonsurgical cases [30]. Therefore, this retrospective study aimed to develop ML classification models using preoperative clinical features and radiomics to differentiate Category 1 from Category 2 patients. These expected classification models will contribute to the development of an ML-based clinical decision support system to facilitate the selection of the appropriate initial management for each individual patient with PMT.”
- In the introduction, what key theoretical perspectives and empirical findings in the main literature have already informed the problem formulation? What major, unaddressed puzzle, controversy, or paradox does this research address?
- Why does it need to be addressed?
- Why it should be now - not in the past?
Author response: Thank you for your critical comments. The Introduction section has been substantially revised for clarity (please see the response to the first comment).
- Further, in the introduction, what is the recent knowledge gap of the main literature that the author needs to write this research? What we have known and what we have not known? What is missing from current works? Please explain and give examples!
Author response: Thank you for your critical comments. The Introduction section has been substantially revised for clarity (please see the response to the first comment).
- Why authors focused on the prevascular mediastinal tumor?
Author response: Thank you for your comment. Since pathological diagnosis is the gold standard diagnostic method, patients with suspected cancer usually undergo biopsy for pathological examination prior to definitive therapy. But, the initial management of patients with suspected PMTs is determined based on clinical diagnosis, but not pathological diagnosis. In order to improve the accuracy of clinical diagnosis, this study aimed to develop ML classification models as a clinical decision support system to facilitate the selection of the initial management for patients with PMTs.
The prevalence, heterogeneous nature, and the selection of the intimal management of PMTs have been articulated in the revised Introduction section.
- Below papers have some interesting implications that you could discuss in your introduction and how it relates to your work.
- Vulli A, Srinivasu PN, Sashank MSK, Shafi J, Choi J, Ijaz MF. Fine-Tuned DenseNet-169 for Breast Cancer Metastasis Prediction Using FastAI and 1-Cycle Policy. Sensors (Basel). 2022 Apr 13;22(8):2988. doi: 10.3390/s22082988. PMID: 35458972; PMCID: PMC9025766.
- Rao, G. E., Rajitha, B., Srinivasu, P. N., Ijaz, M. F., & Woźniak, M. (2024). Hybrid framework for respiratory lung diseases detection based on classical CNN and quantum classifiers from chest X-rays. Biomedical Signal Processing and Control, 88, 105567.
- Praveen SP, Srinivasu PN, Shafi J, Wozniak M, Ijaz MF. ResNet-32 and FastAI for diagnoses of ductal carcinoma from 2D tissue slides. Sci Rep. 2022 Dec 2;12(1):20804. doi: 10.1038/s41598-022-25089-2. PMID: 36460697; PMCID: PMC9716161.
- Ahmad I, Merla A, Ali F, Shah B, AlZubi AA, AlZubi MA. A deep transfer learning approach for COVID-19 detection and exploring a sense of belonging with Diabetes. Front Public Health. 2023 Nov 6;11:1308404. doi: 10.3389/fpubh.2023.1308404. PMID: 38026271; PMCID: PMC10657998.
Author response: Thank you for your comment. Those important references were cited in the Introduction section as follows:
“With the advancement of machining learning (ML) technology, various image-based ML models have been developed to predict breast cancer metastasis [20], lung diseases [21,22], ductal carcinoma [23].”
- It would be interesting if the authors report the trade-off compared to other methods especially the computational complexity of the models. Some techniques require more memory space and take longer time, please elaborate on that.
Author response: Thank you for your comment. The performance of 3D CNN model was briefly compared with that of ML model as follows:
“Deep learning utilizes artificial neural networks to simulate human brain. Generally speaking, deep learning models outperformed the ML models; however, deep learning models are more complex, and require more parameters, huge amounts of data, and high computational cost. In our recent study, we found that the radiomics-based ML model had superior performance compared to the 3D conventional neural network (CNN) model within a limited dataset [29]. Due to the limited sample size, the present study focused on the development of ML models using both clinical features and radiomics.”
- What are the practical implications of your research?
Author response: The potential clinical application of the findings was discussed as follows:
“Compared with other clinical-radiomics model studies that limited specific PMT histology types [27-29], this study did not set restrictions on PMT subtypes, including12 different PMT subtypes. In addition, unlike other clinical-radiomics model studies aimed to improve differential diagnosis [27-29], the developed voting ensemble ML model may be utilized as a clinical decision support system to help the selection of initial management for patients with PMTs.”
- What are the limitations of the present work?
Author response: The limitations of this study were articulated as follows:
“Some limitations need to be addressed. First of all, the findings of this single-institution retrospective study need to be confirmed by other studies conducted in distinct geographic areas. In addition, this retrospective study did not have a training dataset and a separated validation dataset because of a spectrum of many PMT subtypes. Instead, 10-fold cross-validation was utilized to train and validate ML models. The mean classification failure rates of ensemble learning models could be calculated in the present retrospective study; however, patients at high risk of misclassification and underlying reasons could not be investigated. Moreover, due to the etiologic diversity of PMTs [6,56], many PMT subtypes are extremely rare and may not be considered in the binary classification system for initial treatment. Therefore, large-scale multicenter studies are warranted to evaluate the feasibility of ensemble ML classification model as a clinical decision support system to facilitate the selection of initial treatment for patients with PMTs.
Furthermore, some unresectable TET cases may receive neoadjuvant chemotherapy first. Once the tumor size is reduced, surgical resection will be performed to improve proportion of R0 resection and the prognosis. However, due to the small number of patients undergoing neoadjuvant chemotherapy followed by surgery, large-scale multicenter studies are needed to develop the clinical-radiomics ML classification model for predicting patients with TET who may undergo neoadjuvant chemotherapy followed by surgical resection.”
- Conclusion is too short. Add more explanation.
Author response: The conclusion was expanded as follows:
“In conclusion, based on preoperative clinical and radiomic features, we developed a voting ensemble learning classification model to discriminate patients with PMT who need direct surgery from those who need CNB, achieving a classification accuracy of 90.4% that was higher than clinical diagnosis (86.1%). This voting ensemble learning clinical-radiomic classification model can serve as a clinical decision support system to assist clinicians to select the appropriate initial treatment for PMT patients to reduce unnecessary CNB and surgical resection and the harm to patients.”
Reviewer 4 Report
Comments and Suggestions for Authors
This is a retrospective study using CECT and clinical data. Final conclusion is this system is useful to determine surgery or biopsy. Both surgery and biopsy were done to determine pathology. Because of its retrospective nature, this study could not answer the need for these procedures. In clinical situations, a common question is whether we need a biopsy or surgery. Neo-Adjuvant chemotherapy is a common therapeutic option, but this study could not answer this question, but author should discuss this issue.
Author Response
Reviewer 4:
1. This is a retrospective study using CECT and clinical data. Final conclusion is this system is useful to determine surgery or biopsy. Both surgery and biopsy were done to determine pathology. Because of its retrospective nature, this study could not answer the need for these procedures. In clinical situations, a common question is whether we need a biopsy or surgery. Neo-Adjuvant chemotherapy is a common therapeutic option, but this study could not answer this question, but author should discuss this issue.
Author response: Thank you for your insightful comment. Among PMTs, lymphoma and malignant germ cell tumors are treated with chemotherapy and/or radiation therapy. In contrast, surgical resection is the preferred treatment for thymoma and thymic carcinoma. Some unresectable TET cases may receive neoadjuvant chemotherapy first. Once the tumor size is reduced, surgical resection will be performed to minimize the risk of surgery and to improve proportion of R0 resection, thereby improving the prognosis. However, due to the small number of patients undergoing neoadjuvant chemotherapy followed by surgery, large-scale multicenter studies are warranted to develop the clinical-radiomics ML classification model for predicting patients with TET who may undergo neoadjuvant chemotherapy followed by surgical resection. The lack of investigation on neoadjuvant chemotherapy was acknowledged as a limitation of this study as follows:
“Furthermore, some unresectable TET cases may receive neoadjuvant chemotherapy first. Once the tumor size is reduced, surgical resection will be performed to improve proportion of R0 resection and the prognosis. However, due to the small number of patients undergoing neoadjuvant chemotherapy followed by surgery, large-scale multicenter studies are needed to develop the clinical-radiomics ML classification model for predicting patients with TET who may undergo neoadjuvant chemotherapy followed by surgical resection.”
Round 2
Reviewer 1 Report
Comments and Suggestions for Authors
The authors improved their manuscript but unfortunately the presentation of the applied methods does not support the replication of the study.
Simple summary
- "more urgent ant for patients with PMTs " please define ant.
Abstract
- " which significantly outperformed clinical diagnosis " please provide the p-value to support this statement.
- Do nt use short forms in scientific writing (e.g. it's, can't etc.).
- "A misclassification rate of 13.9% was noted in the current retrospective study, which resulted " it it is a result of your current study, this is a results. Otherwise reference(s) is/are needed.
- Please do not use meanless expressions such as "recently". Please be specific in your writing.
- Since previously ML models were reported, please clearly specify which are the limits of these reported ML classification algorithms.
- Please define category 1 and 2 in the aim of your study.
Methods
- Please define "without complete contrast-enhanced computed tomography (CECT) data "
- Please provide the date of ethics approval (dd/mm/yyy).
- Do not start a sentence with an abbreviation.
- Briefly present the significance of the extracted radiomic features.
- Define optimal lambda values.
- Do not use abbreviation in subtitles.
- How the build in algorithm were chosen to be used in your study?
- How resulted exactly 14 models?
- You must adjust the significance level because the number of radioman feature is high the the chance identification is also high.
Results
- How many patients were in Category 2 after exclusion of those with missing data?
- "Category 2 had significantly more patients with abnormal levels of LDH " please also report metrics of centrality and dispersion for these variables along with the number of patients investigated.
- Delete the "%" symbol from the body of the table. This information must stay in the definition of the rows.
- Table 1: abnormal means higher or smaller values.
- The same results are in text and tables.
- The main issue related with the high number of radionics feature per patients (means a high risk of chance association) is not consider by the authors as a limitation of their study. The number of factors in higher than the number of patients in each category and this must be appropriately discussed.
Author Response
The Reviewer
The authors improved their manuscript but unfortunately the presentation of the applied methods does not support the replication of the study.
Simple summary
- "more urgent ant for patients with PMTs " please define ant.
Author response: Thank you for pointing it out. The typo was deleted in the revised text.
Abstract
- " which significantly outperformed clinical diagnosis " please provide the p-value to support this statement.
Author response: Thank you for your comment. The p-value was added accordingly.
- Do not use short forms in scientific writing (e.g. it's, can't etc.).
Author response: Thank you for pointing it out. The short form was corrected in the revised text.
- "A misclassification rate of 13.9% was noted in the current retrospective study, which resulted " it it is a result of your current study, this is a results. Otherwise reference(s) is/are needed.
Author response: Thank you for your valuable comment. The corresponding statement was revised as follows:
“The accuracy of such binary classification was validated by the final pathology report, and the misclassification resulted in unnecessary CNB and surgical resection with a negative impact on patient health.”
- Please do not use meaningless expressions such as "recently". Please be specific in your writing.
Author response: Thank you for pointing it out. “Recently” was deleted.
- Since previously ML models were reported, please clearly specify which are the limits of these reported ML classification algorithms.
Author response: Thank you for your valuable comment. The previously developed ML models focused on differential diagnosis between certain PMT subtypes. No ML models predicting the selection of the initial management between direct surgery and CNB have been built. The corresponding statements were revised for clarity as follows:
“Several clinical-radiomics models have been developed to predict pathological subtypes of PMTs [27], to differentiate low-risk from high-risk thymomas [28], and to distingue TETs from other PMT subtypes [29], thereby facilitating clinical diagnosis. However, no clinical-radiomics models have been developed to predict the selection of the initial management between direct surgery and CNB.”
- Please define category 1 and 2 in the aim of your study.
Author response: Thank you for you valuable comment. The study aim was revised for clarity as follows:
“Therefore, this retrospective study aimed to develop ML classification models using preoperative clinical features and radiomics to differentiate Category 1 patients who need immediate surgical resection from Category 2 patients who need CNB first.”
Methods
- Please define "without complete contrast-enhanced computed tomography (CECT) data"
Author response: In patients received CECT of the head and neck, the entire chest was not completely scanned, and only the upper mediastinum was the focus of CECT scan. The corresponding statement was revised for clarity as follows:
“We enrolled patients who underwent a complete contrast-enhanced computed tomography (CECT) of the chest. Patients who underwent non-contrast-enhanced computed tomography alone or only had head and neck computed tomography without complete coverage of the entire thoracic cavity were excluded.”
- Please provide the date of ethics approval (dd/mm/yyy).
Author response: The data of ethics approval was 04/10/2022, which was added into the revised text.
- Do not start a sentence with an abbreviation.
Author response: Thank you for pointing it out, which was corrected.
- Briefly present the significance of the extracted radiomic features.
Author response: The extracted radiomic features reflect subtle characteristics of MPTs on images, such as sphericity, diameter, homogeneity, and calcification. This information was added to the subsection of 2.4. Radiomic feature extraction and selection.
- Define optimal lambda values.
Author response: The optimal lambda is defined as the lambda value at which the machine learning model had the best predictive performance. In this study, the CatBoost model with the third LASSO selection (Selection_3) had a highest accuracy. Therefore, the optimal lambda was 0.025354 in Selection_3. This information was added to the subsection of 3.4. The individual machine learning model as follows:
“The CatBoost model with the third LASSO selection (Selection_3) had the best classification performance, with the optimal lambda of 0.025354, a Ln (lambda) of -3.675, and an accuracy of 0.8422 ± 0.0423 (Table 3).”
- Do not use abbreviation in subtitles.
Author response: Thank you for pointing it out. The subtitles throughout the text were corrected.
- How the build in algorithm were chosen to be used in your study?
- How resulted exactly 14 models?
Author response: These 14 algorithms are the default algorithms of the predicator learderboard function of AutoGluon Tabular classifier. We did not try custom algorithms, because many default algorithms have been commonly used to develop classification models.
The corresponding statement in the subsection of 2.5. Machine learning model building was revised as follows:
“Python version 3.8.9 with AutoGluon Tubular classifier version 0.8.2 [34], was used to model 14 default ML algorithms, including CatBoost, ExtraTrees with Entropy, ExtraTrees with Gini, Kneighbors with Distance Weights, Kneighbors with Uniform Weights, LightGBM, LightGBMLarge, LightGBM with ExtraTress, NeurlNetFastAI, NeuralNetTorch, RandomForest with Entropy, RandomForest with Gini, WeightedEnsemble_L2, and XGBoost to develop classification models.”
In addition, the common utilization of default algorithms has been discussed in the Discussion section as follows:
“ML algorithms used in this study have been commonly utilized to develop classification models for various diseases [49-53]. Ensemble learning techniques have been widely used in clinical practice to combine several ML models to create a stronger model with superior performance. The ensemble model has a better predictive accuracy than those of individual models [50]. RandomForest, CatBoost and XGBoost were used to develop an ensemble model for predicting pre-cancer in pre- and post-menopausal women, with an accuracy of 94% [51].”
- You must adjust the significance level because the number of radioman feature is high the the chance identification is also high.
Author response: Thank you for your critical comment. However, based on the following reasons, statistical significance was set at a p-value of 0.05 or 95% conference interval (95% CI).
- Bonferroni correction may be used to adjust the significance level, for example, 0.05/(851+8 features) = 0.0000582. However, Bonferroni correction is a relatively conservative, familywise error rate control method and will increase the probability of type II error (false negative).
- If Mann-Whitney U test was used to compare all variables with Categories 1 and 2, a large number of variables with p value <0.05 will be identified. However, in the present study, variable selection was not based on p value. LASSO regression was used for variable selection, instead. Only 22 variables were selected in Selection_3, so the number of selected features was no much larger than the number of patients.
Results
- How many patients were in Category 2 after exclusion of those with missing data?
Author response: There were 68 patients in Category 2 with all data on the serum levels of LDH, AFP, and HCG, as revealed in Table 1.
- "Category 2 had significantly more patients with abnormal levels of LDH " please also report metrics of centrality and dispersion for these variables along with the number of patients investigated.
Author response: Thank you for your valuable comment. The metrics of centrality and dispersion for these variables along with the number of patients are presented as Supplementary Table S1. The previous Supplementary Table S1 was changed to Supplementary Table S2.
|
Supplementary Table S1. The centrality and dispersion for age, LDH, AFP and HCG. |
|||||
|
|
Age |
LDH |
AFP |
HCG |
|
|
Category 1 patients |
N |
182 |
31 |
45 |
31 |
|
Mean |
60.02 |
210.10 |
2.46833 |
.40555 |
|
|
Std. Deviation |
15.629 |
91.214 |
1.299046 |
.905478 |
|
|
Grouped Median |
60.87 |
189.00 |
2.29000 |
.17167 |
|
|
Kurtosis |
.292 |
4.856 |
1.140 |
6.957 |
|
|
Skewness |
-.463 |
1.920 |
.766 |
2.679 |
|
|
Range |
87 |
423 |
6.470 |
3.700 |
|
|
|
|
|
|
|
|
|
Category 2 patients |
N |
193 |
116 |
82 |
68 |
|
Mean |
52.70 |
392.15 |
1128.34102 |
2994.00781 |
|
|
Std. Deviation |
20.945 |
376.722 |
6176.807435 |
21326.820646 |
|
|
Grouped Median |
54.50 |
250.50 |
2.10500 |
.12727 |
|
|
Kurtosis |
-1.054 |
6.735 |
53.231 |
65.321 |
|
|
Skewness |
.017 |
2.537 |
7.009 |
8.025 |
|
|
Range |
83 |
1979 |
50610.000 |
174608.000 |
|
|
|
|
|
|
|
|
|
Total patient |
N |
375 |
147 |
127 |
99 |
|
Mean |
56.25 |
353.76 |
729.40976 |
2056.61720 |
|
|
Std. Deviation |
18.890 |
345.034 |
4981.887025 |
17689.109293 |
|
|
Grouped Median |
59.14 |
238.00 |
2.21667 |
.14103 |
|
|
Kurtosis |
-.649 |
9.078 |
83.142 |
95.156 |
|
|
Skewness |
-.273 |
2.884 |
8.752 |
9.685 |
|
|
Range |
92 |
1979 |
50610.000 |
174608.000 |
|
|
|
|
|
|
|
|
- Delete the "%" symbol from the body of the table. This information must stay in the definition of the rows.
Author response: Thank you for your comment. Table 1 has been corrected accordingly.
- Table 1: abnormal means higher or smaller values.
Author response: Thank you for your comment. “Abnormal” has been replaced with “higher” in Table 1 and throughout the revised text.
- The same results are in text and tables.
Author response: Thank you for pointing it out. To reduce unnecessary repetition, only accuracy was presented in the text.
- The main issue related with the high number of radionics feature per patients (means a high risk of chance association) is not consider by the authors as a limitation of their study. The number of factors in higher than the number of patients in each category and this must be appropriately discussed.
Author response: Thank you for your critical comment. However, the number of factors is not higher than the number of patients in the present study design.
- a) Bonferroni correction may be used to adjust the significance level, for example, 0.05/(851+8 features) = 0.0000582. However, Bonferroni correction is a relatively conservative, familywise error rate control method and will increase the probability of type II error (false negative).
- If Mann-Whitney U test was used to compare all variables with Categories 1 and 2, a large number of variables with p value <0.05 will be identified. However, in the present study, variable selection was not based on p value. LASSO regression was used for variable selection, instead. Only 22 variables were selected in Selection_3, so the number of selected features was no much larger than the number of patients.
Reviewer 2 Report
Comments and Suggestions for Authors
My previous comments have been addressed.
Author Response
Samuel C. Mok, Ph.D.
Editor-in-Chief,
Cancers
RE: cancer-27257955R1
Dear Prof. Mok,
We deeply appreciate you and the reviewers for the valuable comments regarding our revised manuscript. The entire manuscript has been thoroughly revised accordingly. The metrics of centrality and dispersion for these variables along with the number of patients are presented as Supplementary Table S1. However, the number of factors is not higher than the number of patients in the present study design. It is therefore reasonable to set statistical significance at a p-value of 0.05 or 95% conference interval.
Below please find our point-by-point response to the comments. The track changes function was used to record modifications made to the manuscript. We are grateful for this opportunity to revise our report and to have it reconsidered for possible publication in Cancers.
Best regards,
Mi-Chia, Ma, Ph.D.
Department of Statistics and Institute of Data Science, National Cheng Kung University, Tainan, Taiwan
Tel: 886-06-2757575-53639
mcma@mail.ncku.edu.tw
Yi-Ting, Yen, Ph.D.
Division of Thoracic Surgery, Department of Surgery, National Cheng Kung University Hospital, College of Medicine, National Cheng Kung University, Tainan, Taiwan
b85401067@gmail.com
Reviewer 3 Report
Comments and Suggestions for Authors.
Author Response

(The authors gave the same response as above.)

Reviewer 4 Report
Comments and Suggestions for Authors
I recognized significant improvement of this manuscript.
Author Response

(The authors gave the same response as above.)

Round 3
Reviewer 1 Report
Comments and Suggestions for Authors
Thank you for your work to improve your manuscript.
Please explain in the statistical analysis why a significance level of 0.05 would be sufficient.
Author Response
Thank you for your feedback and suggestion on the correct interpretation of p-values. The p-value represents the probability of committing a Type I error, while the significance level denotes the upper limit of acceptable Type I error probability, typically set at 0.05, 0.01, or 0.001. The choice of significance level depends on the researcher's willingness to take on the risk of making a decisional error. A Type I error occurs when the null hypothesis is true but rejected. In our study, it means falsely predicting the relevance of model features and intercepts when they are not.
Setting a significance level of 0.05 means allowing the possibility (or probability) of a false alarm to occur, which should be less than 0.05 (i.e., only 1 occurrence in 20). A significance level of 0.05 corresponds to a confidence level of 0.95, where the confidence interval represents the non-rejection region of the test. When calculating the confidence interval for accuracy, it's akin to conducting a test on accuracy. The approach of 10-fold cross-validation involves conducting 10 tests on the sample. Therefore, calculating the 95% confidence interval for these 10 tests (mean ± (standard deviation * 1.96 / square root of 10)), if the confidence interval encompasses 0.861, it implies no significant difference between the accuracy of the machine learning model and clinical diagnosis (accuracy=0.861). The accuracy of the machine learning model proposed in our study is 0.9044 even with 99% confidence interval between 0.871 and 0.938, higher than the current clinical prediction accuracy as 0.861.
Sentences explaining the significance level were added in the statistical analysis as 【The p-value represents the probability of committing a Type I error, while the significance level denotes the upper limit of acceptable Type I error probability, typically set at 0.05, 0.01, or 0.001. The choice of significance level depends on the researcher's willingness to take on the risk of making a decisional error. Setting a significance level of 0.05 means allowing the possibility (or probability) of a false alarm to occur, which should be less than 0.05 (i.e., only 1 occurrence in 20).】